# Emergent Correspondence from Image Diffusion

Luming Tang[*]      Menglin Jia[*]      Qianqian Wang[*]
Cheng Perng Phoo      Bharath Hariharan
Cornell University

## Abstract

Finding correspondences between images is a fundamental problem in computer vision. In this paper, we show that correspondence emerges in image diffusion models *without any explicit supervision*. We propose a simple strategy to extract this implicit knowledge out of diffusion networks as image features, namely DIffusion FeaTures (DIFT), and use them to establish correspondences between real images. Without any additional fine-tuning or supervision on the task-specific data or annotations, DIFT is able to outperform both weakly-supervised methods and competitive off-the-shelf features in identifying semantic, geometric, and temporal correspondences. Particularly for semantic correspondence, DIFT from Stable Diffusion is able to outperform DINO and OpenCLIP by 19 and 14 accuracy points respectively on the challenging SPair-71k benchmark. It even outperforms the state-of-the-art supervised methods on 9 out of 18 categories while remaining on par for the overall performance. Project page: https://diffusionfeatures.github.io.

## 1 Introduction

Drawing correspondences between images is a critical primitive in 3D reconstruction [73], object tracking [22, 90], video segmentation [92], image and video editing [102, 58, 98]. This problem of drawing correspondence is easy for humans: we can match object parts not only across different viewpoints, articulations and lighting changes, but even across drastically different categories (e.g., between cats and horses) or different modalities (e.g., between photos and cartoons). Yet, we rarely if ever get explicit correspondence labels for training. The question is, can computer vision systems similarly learn accurate correspondences without any labeled data at all?

There is indeed some evidence that contrastive self-supervised learning techniques produce good correspondences as a side product of learning on unlabeled data [10, 28]. However, in this paper, we look to a new class of self-supervised models that has been attracting attention: diffusion-based generative models [32, 79]. While diffusion models are primarily models for image synthesis, a key observation is that these models produce good results for image-to-image translation [53, 85] and image editing [8, 80]. For instance, they can convert a dog to a cat without changing its pose or context [61]. It would appear that to perform such editing, the model must implicitly reason about correspondence between the two categories (e.g., the model needs to know where the dog's eye is in order to replace it with the cat's eye). We therefore ask, do image diffusion models learn correspondences?

We answer the question in the affirmative by construction: we provide a simple way of extracting correspondences on real images using pre-trained diffusion models. These diffusion models [41] have at the core a U-Net [71, 17, 70] that takes noisy images as input and produces clean images as output. As such they already extract features from the input image that can be used for correspondence. Unfortunately, the U-Net is trained to *de-noise*, and so has been trained on *noisy* images. Our strategy

---

[*]Equal contribution.

37th Conference on Neural Information Processing Systems (NeurIPS 2023).

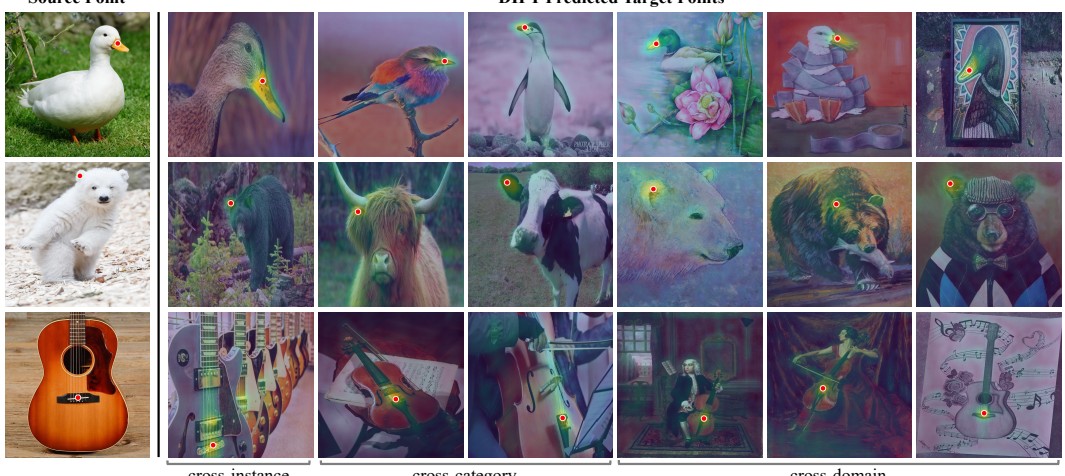

**Source Point**
**DIFT Predicted Target Points**

cross-instance      cross-category      cross-domain

Figure 1: Given a red source point in an image (far left), we would like to develop a model that automatically finds the corresponding point in the images on the right. Without any fine-tuning or correspondence supervision, our proposed diffusion features (DIFT) could establish semantic correspondence across instances, categories and even domains, e.g., from a duck to a penguin, from a photo to an oil-painting. More results are in Figs. 15 and 16 of Appendix E.

for handling this issue is simple but effective: we *add noise* to the input image (thus simulating the forward diffusion process) before passing it into the U-Net to extract feature maps. We call these feature maps (and through a slight abuse of notation, our approach) **DI**ffusion **F**ea**T**ures (**DIFT**). DIFT can then be used to find matching pixel locations in the two images by doing simple nearest neighbor lookup using cosine distance. We find the resulting correspondences are surprisingly robust and accurate (Fig. 1), even across multiple categories and image modalities.

We evaluate DIFT with two different types of diffusion models, on three groups of visual correspondence tasks including semantic correspondence, geometric correspondence, and temporal correspondence. We compare DIFT with other baselines, including task-specific methods, and other self-supervised models trained with similar datasets and similar amount of supervision (DINO [10] and OpenCLIP [36]). Although simple, DIFT demonstrates strong performance on all tasks without any additional fine-tuning or supervision, outperforms both weakly-supervised methods and other self-supervised features, and even remains on par with the state-of-the-art supervised methods on semantic correspondence.

## 2   Related Work

**Visual Correspondence.** Establishing visual correspondences between different images is crucial for various computer vision tasks such as Structure-from-Motion / 3D reconstruction [2, 73, 60, 74], object tracking [22, 97], image recognition [63, 81, 9] and segmentation [50, 47, 72, 28]. Traditionally, correspondences are established using hand-designed features, such as SIFT [51] and SURF [6]. With the advent of deep learning, methods that learn to find correspondences in a supervised-learning regime have shown promising results [46, 14, 42, 35]. However, these approaches are difficult to scale due to the reliance on ground-truth correspondence annotations. To overcome difficulties in collecting a large number of image pairs with annotated correspondences, recent works have started looking into how to build visual correspondence models with weak supervision [91] or self-supervision [92, 37]. Meanwhile, recent works on self-supervised representation learning [10] has yielded strong per-pixel features that could be used to identify visual correspondence [84, 3, 10, 28]. In particular, recent work has also found that the internal representation of Generative Adversarial Networks (GAN) [23] could be used for identifying visual correspondence [99, 62, 57] within certain image categories. Our work shares similar spirits with these works: we show that diffusion models could generate features that are useful for identifying visual correspondence on general images. In addition, we show that features generated at different timesteps and different layers of the de-noising process

encode different information that could be used for determining correspondences needed for different downstream tasks.

**Diffusion Model** [78, 32, 79, 41] is a powerful family of generative models. Ablated Diffusion Model [17] first showed that diffusion could surpass GAN's image generation quality on ImageNet [15]. Subsequently, the introduction of classifier-free guidance [33] and latent diffusion model [70] made it scale up to billions of text-image pairs [75], leading to the popular open-sourced text-to-image diffusion model, i.e., Stable Diffusion. With its superior generation ability, recently people also start investigating the internal representation of diffusion models. For example, previous works [85, 31] found that the intermediate-layer features and attention maps of diffusion models are crucial for controllable generations; other works [5, 94, 101] explored adapting pre-trained diffusion models for various downstream visual recognition tasks. Different from these works, we are the first to directly evaluate the efficacy of features inherent to pre-trained diffusion models on various visual correspondence tasks.

# 3 Problem Setup

Given two images $I_1, I_2$ and a pixel location $p_1$ in $I_1$, we are interested in finding its corresponding pixel location $p_2$ in $I_2$. Relationships between $p_1$ and $p_2$ could be semantic correspondence (i.e., pixels of different objects that share similar semantic meanings), geometric correspondence (i.e., pixels of the same object captured from different viewpoints), or temporal correspondence (i.e., pixels of the same object in a video that may deform over time).

The most straightforward approach to obtaining pixel correspondences is to first extract dense image features in both images and then match them. Specifically, given a feature map $F_i$ for image $I_i$, we can extract a feature vector $F_i(p)$ for pixel location $p$ through bilinear interpolation. Then given a pixel $p_1$ in image $I_1$, we can obtain the corresponding pixel in image $I_2$ as:

$$p_2 = \arg \min_p d(F_1(p_1), F_2(p)) \tag{1}$$

where $d$ is a distance metric and we use cosine distance by default in this work.

# 4 Diffusion Features (DIFT)

In this section, we first review what diffusion models are and then explain how we extract dense features on real images using pre-trained diffusion models.

## 4.1 Image Diffusion Model

Diffusion models [32, 79] are generative models that aim to transform a Normal distribution to an arbitrary data distribution. In our case, we use image diffusion models, thus the data distribution and the Gaussian prior are both over the space of 2D images.

During training, Gaussian noise of different magnitudes is added to clean data points to obtain noisy data points. This is typically thought of as a "diffusion" process, where the starting point of the diffusion $x_0$ is a clean image from the training dataset and $x_t$ is a noisy image obtained by "mixing" $x_0$ with noise:

$$x_t = \sqrt{\alpha_t} x_0 + (\sqrt{1 - \alpha_t}) \epsilon \tag{2}$$

where $\epsilon \sim \mathcal{N}(0, \mathbf{I})$ is the randomly-sampled noise, and $t \in [0, T]$ indexes "time" in the diffusion process with larger time steps involving more noise. The amount of noise is determined by $\{\alpha_t\}_1^T$, which is a pre-defined noise schedule. We call this the diffusion *forward* process.

A neural network $f_\theta$ is trained to take $x_t$ and time step $t$ as input and predict the input noise $\epsilon$. For image generation, $f_\theta$ is usually parametrized as a U-Net [71, 17, 70]. Once trained, $f_\theta$ can be used to "reverse" the diffusion process. Starting from pure noise $x_T$ sampled from a Normal distribution, $f_\theta$ can be iteratively used to estimate noise $\epsilon$ from the noisy data $x_t$ and remove this noise to get a cleaner data $x_{t-1}$, eventually leading to a sample $x_0$ from the original data distribution. We call this the diffusion *backward* process.

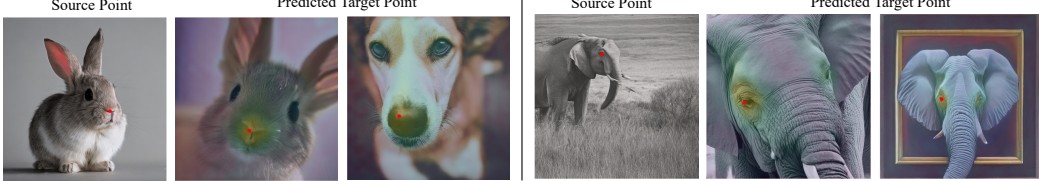

Figure 2: Given a Stable Diffusion generated image, we extract its intermediate layer activations at a certain time step $t$ during its backward process, and use them as the feature map to predict the corresponding points. Although simple, this method produces correct correspondences on generated images already not only within category, but also cross-category, even in cross-domain situations, e.g., from a photo to an oil painting.

## 4.2 Extracting Diffusion Features on Real Images

We hypothesize that diffusion models learn correspondence implicitly [85, 61] in Sec. 1, but how can we extract this correspondence? Consider first *generated* images, where we have access to the complete internal state of the network throughout the entire backward process. Given a generated image from Stable Diffusion [70] , we extract the feature maps of its intermediate layers at a specific time step $t$ during the backward process, which we then utilize to establish correspondences between two different generated images as described in Sec. 3. As illustrated in Fig. 2, this straightforward approach allows us to find correct correspondences between generated images, even when they belong to different categories or domains.

Replicating this approach for real images is challenging because of the fact that the real image itself does not belong to the training distribution of the U-Net (which was trained on noisy images), and we do not have access to the intermediate noisy images that would have been produced during the generation of this image. Fortunately, we found a simple approximation using the forward diffusion process to be effective enough. Specifically, we first add *noise* of time step $t$ to the real image (Eq. (2)) to move it to the $x_t$ distribution, and then feed it to network $f_\theta$ together with $t$ to extract the intermediate layer activations as our DIffusion FeaTures, namely DIFT. As shown in Figs. 1 and 3, this approach yields surprisingly good correspondences for real images.

Moving forward, a crucial consideration is the selection of the time step $t$ and the network layer from which we extract features. Intuitively we find that a larger $t$ and an earlier network layer tend to yield more semantically-aware features, while a smaller $t$ and a later layer focus more on low-level details. The optimal choices of $t$ and layer depend on the specific correspondence task at hand, as different tasks may require varying trade-offs between semantic and low-level features. For example, semantic correspondence likely benefits from more semantic-level features, whereas geometric correspondence between two views of the same instance may perform well with low-level features. We therefore use a 2D grid search to determine these two hyper-parameters for each correspondence task. For a comprehensive list of the hyper-parameter values used in this paper, please refer to Appendix C.

Lastly, to enhance the stability of the representation in the presence of random noise added to the input image, we extract features from multiple noisy versions with different samples of noise, and average them to form the final representation.

## 5 Semantic Correspondence

In this section, we investigate how to use DIFT to identify pixels that share similar semantic meanings across images, e.g., the eyes of two different cats in two different images.

### 5.1 Model Variants and Baselines

We extract DIFT from two commonly used, open-sourced image diffusion models: Stable Diffusion 2-1 (SD) [70] and Ablated Diffusion Model (ADM) [17]. SD is trained on the LAION [75] whereas ADM is trained on ImageNet [15] without labels. We call these two features $\text{DIFT}_{sd}$ and $\text{DIFT}_{adm}$ respectively.

To separate the impact of training data on the performance of DIFT, we also evaluate two other commonly used self-supervised features as baselines that share basically the same training data:

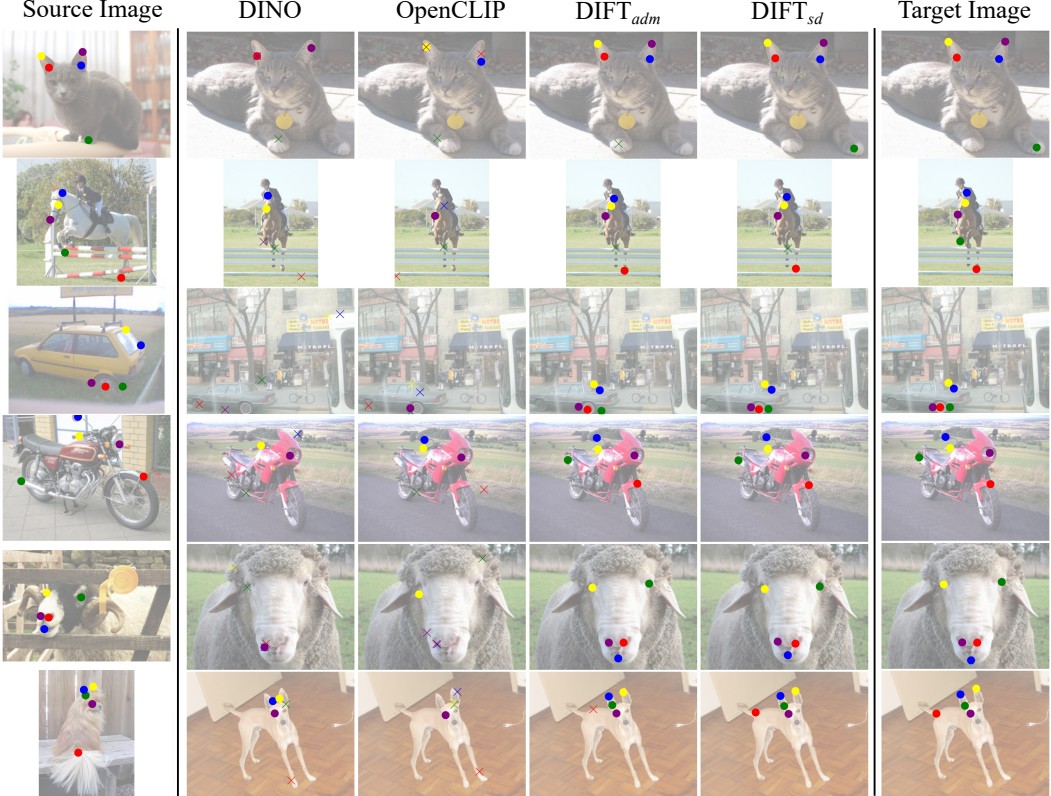

| Source Image | DINO | OpenCLIP | $\text{DIFT}_{adm}$ | $\text{DIFT}_{sd}$ | Target Image |

Figure 3: Visualization of semantic correspondence prediction on SPair-71k using different features. The leftmost image is the source image with a set of keypoints; the rightmost image contains the ground-truth correspondence for a target image whereas any images in between contain keypoints found using feature matching with various features. Different colors indicate different keypoints. We use circles to indicate correctly-predicted points under the threshold $\alpha_{bbox} = 0.1$ and crosses for incorrect matches. DIFT is able to establish correct correspondences under clustered scenes (row 3), viewpoint changes (row 2 and 4), and occlusions (row 5). See Fig. 17 in Appendix E for more results.

OpenCLIP [36] with ViT-H/14 [18] trained on LAION, as well as DINO [10] with ViT-B/8 trained on ImageNet [15] without labels. Note that for both DIFT and other self-supervised features, we do not fine-tune or re-train the models with any additional data or supervision.

## 5.2 Benchmark Evaluation

**Datasets.** We conduct evaluation on three popular benchmarks: SPair-71k [55], PF-WILLOW [27] and CUB-200-2011 [89]. SPair-71k is the most challenging semantic correspondence dataset, containing diverse variations in viewpoint and scale with 12,234 image pairs on 18 categories for testing. PF-Willow is a subset of PASCAL VOC dataset [20] with 900 image pairs for testing. For CUB, following [58], we evaluate 14 different splits of CUB (each containing 25 images) and report the average performance across all splits.

**Evaluation Metric.** Following prior work, we report the percentage of correct keypoints (PCK). The predicted keypoint is considered to be correct if they lie within $\alpha \cdot \max(h, w)$ pixels from the ground-truth keypoint for $\alpha \in [0, 1]$, where $h$ and $w$ are the height and width of either the image ($\alpha_{img}$) or the bounding box ($\alpha_{bbox}$). To find a suitable time step and layer feature to use for DIFT and other self-supervised features, we grid search the hyper-parameters using SPair-71k and use the same hyper-parameter settings for PF-WILLOW and CUB.

We observed inconsistencies in PCK measurements across prior literature[1]. Some works [35, 42, 14] use the total number of correctly-predicted points in the whole dataset (or each category split) divided

---

[1] ScorrSAN [35] and GANgealing [62]'s evaluation code snippets, which calculate PCK per image and PCK per point respectively.

Table 1: PCK($\alpha_{bbox} = 0.1$) `per image` on SPair-71k. All the DIFT results have gray background for easy lookups. Methods are grouped into 3 groups: (a) fully supervised with correspondence annotations, (b) weakly supervised with in-domain image collections, (c) no supervision. Best numbers in group (a) are **bolded**. Among groups (b) and (c) taken together, we annotate **best** and second-best results. Without any supervision, both $DIFT_{sd}$ and $DIFT_{adm}$ outperform previous weakly-supervised methods and self-supervised techniques by a large margin.

| Sup. | Method | Aero | Bike | Bird | Boat | Bottle | Bus | Car | Cat | Chair | Cow | Dog | Horse | Motor | Person | Plant | Sheep | Train | TV | All |
|------|--------|------|------|------|------|--------|-----|-----|-----|-------|-----|-----|-------|-------|--------|-------|-------|-------|----|-----|
| (a) | CATs [14] | 52.0 | 34.7 | 72.2 | 34.3 | 49.9 | 57.5 | 43.6 | 66.5 | 24.4 | 63.2 | 56.5 | 52.0 | 42.6 | 41.7 | 43.0 | 33.6 | 72.6 | 58.0 | 49.9 |
| | MMNet [100] | 55.9 | 37.0 | 65.0 | 35.4 | 50.0 | 63.9 | 45.7 | 62.8 | **28.7** | 65.0 | 54.7 | 51.6 | 38.5 | 34.6 | 41.7 | 36.3 | 77.7 | 62.5 | 50.4 |
| | TransforMatcher [42] | **59.2** | 39.3 | 73.0 | **41.2** | **52.5** | **66.3** | **55.4** | 67.1 | 26.1 | 67.1 | 56.6 | **53.2** | 45.0 | 39.9 | 42.1 | 35.3 | 75.2 | 68.6 | 53.7 |
| | SCorrSAN [35] | 57.1 | **40.3** | **78.3** | 38.1 | 51.8 | 57.8 | 47.1 | **67.9** | 25.2 | **71.3** | **63.9** | 49.3 | **45.3** | **49.8** | **48.8** | **40.3** | **77.7** | **69.7** | **55.3** |
| (b) | NCNet [69] | 17.9 | 12.2 | 32.1 | 11.7 | 29.0 | 19.9 | 16.1 | 39.2 | 9.9 | 23.9 | 18.8 | 15.7 | 17.4 | 15.9 | 14.8 | 9.6 | 24.2 | 31.1 | 20.1 |
| | CNNGeo [67] | 23.4 | 16.7 | 40.2 | 14.3 | 36.4 | 27.7 | 26.0 | 32.7 | 12.7 | 27.4 | 22.8 | 13.7 | 20.9 | 21.0 | 17.5 | 10.2 | 30.8 | 34.1 | 20.6 |
| | WeakAlign [68] | 22.2 | 17.6 | 41.9 | 15.1 | 38.1 | 27.4 | 27.2 | 31.8 | 12.8 | 26.8 | 22.6 | 14.2 | 20.0 | 22.2 | 17.9 | 10.4 | 32.2 | 35.1 | 20.9 |
| | A2Net [76] | 22.6 | 18.5 | 42.0 | 16.4 | 37.9 | 30.8 | 26.5 | 35.6 | 13.3 | 29.6 | 24.3 | 16.0 | 21.6 | 22.8 | 20.5 | 13.5 | 31.4 | 36.5 | 22.3 |
| | SFNet [45] | 26.9 | 17.2 | 45.5 | 14.7 | 38.0 | 22.2 | 16.4 | 55.3 | 13.5 | 33.4 | 27.5 | 17.7 | 20.8 | 21.1 | 16.6 | 15.6 | 32.2 | 35.9 | 26.3 |
| | PMD [48] | 26.2 | 18.5 | 48.6 | 15.3 | 38.0 | 21.7 | 17.3 | 51.6 | 13.7 | 34.3 | 25.4 | 18.0 | 20.0 | 24.9 | 15.7 | 16.3 | 31.4 | 38.1 | 26.5 |
| | PSCNet [38] | 28.3 | 17.7 | 45.1 | 15.1 | 37.5 | 30.1 | 27.5 | 47.4 | 14.6 | 32.5 | 26.4 | 17.7 | 24.9 | 24.5 | 19.9 | 16.9 | 34.2 | 37.9 | 27.0 |
| | PWarpC [83] | 37.4 | 28.8 | 60.8 | 22.9 | 40.5 | 29.4 | 22.8 | 60.1 | 19.5 | 37.8 | 38.4 | 27.9 | 32.1 | 29.7 | 29.2 | 20.2 | 44.5 | 50.0 | 35.3 |
| (c) | DINO [10] | 43.6 | 27.2 | 64.9 | 24.0 | 30.5 | 31.4 | 28.3 | 55.2 | 16.8 | 40.2 | 37.1 | 32.9 | 29.1 | 41.1 | 22.0 | 26.8 | 36.4 | 26.9 | 33.9 |
| | $DIFT_{adm}$ (ours) | 49.7 | 39.2 | 77.5 | 29.3 | 40.9 | 36.1 | 30.5 | 75.5 | 23.7 | 63.7 | 52.8 | 49.3 | 34.1 | 52.3 | 39.3 | 37.3 | 59.6 | 45.4 | 46.3 |
| | OpenCLIP [36] | 51.7 | 31.4 | 68.7 | 28.4 | 31.5 | 34.9 | 36.1 | 56.4 | 21.1 | 44.5 | 41.5 | 41.2 | 41.2 | 51.8 | 21.7 | 28.6 | 46.3 | 20.7 | 38.4 |
| | $DIFT_{sd}$ (ours) | 61.2 | 53.2 | 79.5 | 31.2 | 45.3 | 39.8 | 33.3 | 77.8 | 34.7 | 70.1 | 51.5 | 57.2 | 50.6 | 41.4 | 51.9 | 46.0 | 67.6 | 59.5 | 52.9 |

Table 2: PCK($\alpha_{bbox} = 0.1$) `per point` of various methods on SPair-71k. The groups and colors follow Tab. 1. "Mean" denotes the PCK averaged over categories. Same as in Tab. 1, without any supervision, both $DIFT_{sd}$ and $DIFT_{adm}$ outperform previous weakly-supervised methods with a large margin, and also outperform their contrastive-learning counterparts by over 14 points.

| Sup. | Method | Aero | Bike | Bird | Boat | Bottle | Bus | Car | Cat | Chair | Cow | Dog | Horse | Motor | Person | Plant | Sheep | Train | TV | Mean | All |
|------|--------|------|------|------|------|--------|-----|-----|-----|-------|-----|-----|-------|-------|--------|-------|-------|-------|----|------|-----|
| (b) | NBB [1, 26] | 29.5 | 22.7 | 61.9 | 26.5 | 20.6 | 25.4 | 14.1 | 23.7 | 14.2 | 27.6 | 30.0 | 29.1 | 24.7 | 27.4 | 19.1 | 19.3 | 24.4 | 22.6 | 27.4 | - |
| | GANgealing [62] | - | 37.5 | - | - | - | - | - | 67.0 | - | - | 23.1 | - | - | - | - | - | - | 57.9 | - | - |
| | NeuCongeal [58] | - | 29.1 | - | - | - | - | - | 53.3 | - | - | 35.2 | - | - | - | - | - | - | - | - | - |
| | ASIC [26] | 57.9 | 25.2 | 68.1 | 24.7 | 35.4 | 28.4 | 30.9 | 54.8 | 21.6 | 45.0 | 47.2 | 39.9 | 26.2 | 48.8 | 14.5 | 24.5 | 49.0 | 24.6 | 36.9 | - |
| (c) | DINO [10] | 45.0 | 29.5 | 66.3 | 22.8 | 32.1 | 36.3 | 31.7 | 54.8 | 18.7 | 43.1 | 39.2 | 34.9 | 31.0 | 44.3 | 23.1 | 29.4 | 38.4 | 27.1 | 36.0 | 36.7 |
| | $DIFT_{adm}$ (ours) | 51.6 | 40.4 | 77.6 | 30.7 | 43.0 | 47.2 | 42.1 | 74.9 | 26.6 | 67.3 | 55.8 | 52.7 | 36.0 | 55.9 | 46.3 | 45.7 | 62.7 | 47.4 | 50.2 | 52.0 |
| | OpenCLIP [36] | 53.2 | 33.4 | 69.4 | 28.0 | 33.3 | 41.0 | 41.8 | 55.8 | 23.3 | 47.0 | 43.9 | 44.1 | 43.5 | 55.1 | 23.6 | 31.7 | 47.8 | 21.8 | 41.0 | 41.4 |
| | $DIFT_{sd}$ (ours) | 63.5 | 54.5 | 80.8 | 34.5 | 46.2 | 52.7 | 48.3 | 77.7 | 39.0 | 76.0 | 54.9 | 61.3 | 53.3 | 46.0 | 57.8 | 57.1 | 71.1 | 63.4 | 57.7 | 59.5 |

Table 3: Comparison with state-of-the-art methods on PF-WILLOW PCK `per image` (left) and CUB PCK `per point` (right). The groups follow Tab. 1. Colors of numbers indicate the **best**, second-best results. All the DIFT results have gray background for better reference. $DIFT_{sd}$ achieves the best results without any fine-tuning or supervision with in-domain annotations or data.

| Sup. | Method | PCK@$\alpha_{bbox}$ | |
|------|--------|------|------|
| | | $\alpha = 0.05$ | $\alpha = 0.10$ |
| (a) | SCNet [29] | 38.6 | 70.4 |
| | DHPF [56] | 49.5 | 77.6 |
| | PMD [48] | - | 75.6 |
| | CHM [54] | 52.7 | 79.4 |
| | CATs [14] | 50.3 | 79.2 |
| | TransforMatcher [42] | - | 76.0 |
| | SCorrSAN [35] | 54.1 | 80.0 |
| (b) | WarpC [82] | 49.0 | 75.1 |
| | PWarpC [83] | 45.0 | 75.9 |
| | GSF [39] | 49.1 | 78.7 |
| (c) | DINO [10] | 30.8 | 51.1 |
| | $DIFT_{adm}$ (ours) | 46.9 | 67.0 |
| | OpenCLIP [36] | 34.4 | 61.3 |
| | $DIFT_{sd}$ (ours) | **58.1** | **81.2** |

| Sup. | Method | PCK@$\alpha_{img} = 0.1$ |
|------|--------|------|
| (b) | GANgealing [62] | 56.8 |
| | NeuCongeal [58] | 65.6 |
| (c) | DINO [10] | 66.4 |
| | $DIFT_{adm}$ (ours) | 78.0 |
| | OpenCLIP [36] | 67.5 |
| | $DIFT_{sd}$ (ours) | **83.5** |

by the total number of predicted points as the final PCK, while some works [62, 58, 26] first calculate a PCK value for each image and then average it across the dataset (or each category split). We denote the first metric as PCK `per point` and the second as PCK `per image`. We calculate both metrics for DIFT and self-supervised features, and compare them to methods using that metric respectively.

**Quantitative Results.** We report our results in Tabs. 1 to 3. In addition to feature matching using DINO and OpenCLIP, we also report state-of-the-art fully-supervised and weakly-supervised methods in the respective tables for completeness. Across the three datasets, we observe that features learned via diffusion yield more accurate correspondences compared to features learned using contrastive approaches ($DIFT_{sd}$ vs. OpenCLIP, $DIFT_{adm}$ vs. DINO).

Source patch            Top-5 nearest neighbor cross-category target patches predicted by $DIFT_{sd}$

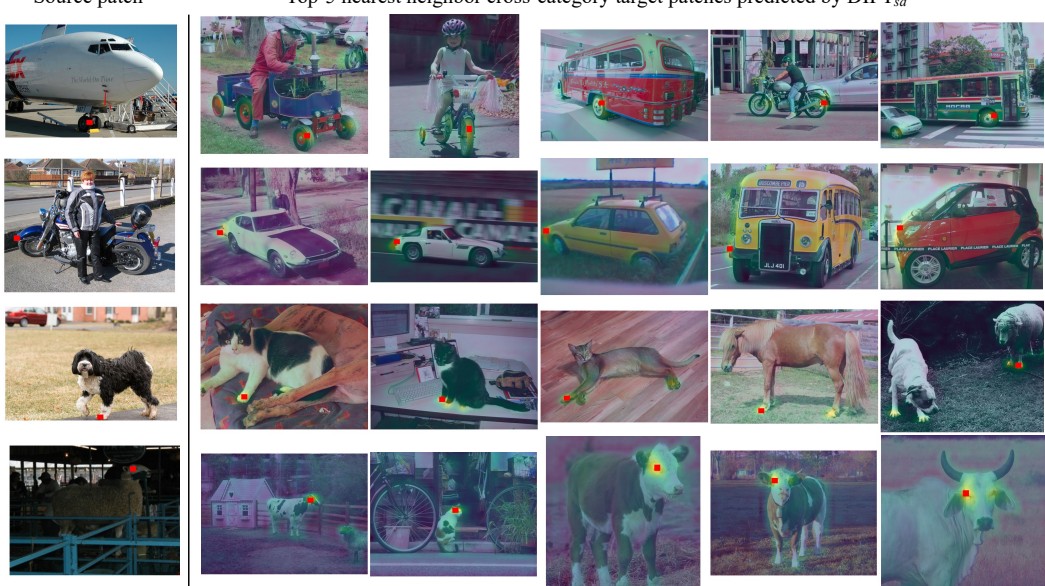

Figure 4: Given image patch specified in the leftmost image (red rectangle), we use $DIFT_{sd}$ to retrieve the top-5 nearest patches in images from different categories in the SPair-71k test set. DIFT is able to find correct correspondence for different objects sharing similar semantic parts, e.g., the wheel of an airplane vs. the wheel of a bus. More results are in Fig. 18 of Appendix E.

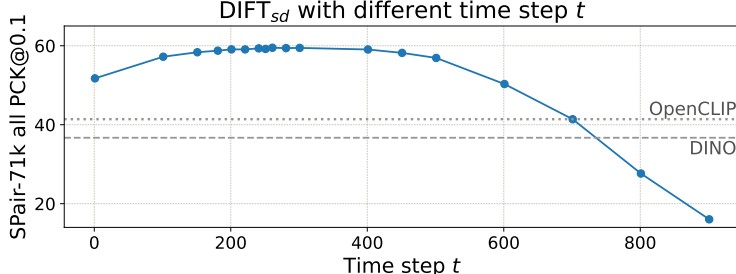

Figure 5: PCK `per point` of $DIFT_{sd}$ on SPair-71k. It maintains high accuracy with a wide range of $t$, outperforming other off-the-shelf self-supervised features.

Furthermore, even without any supervision (be it explicit correspondence or in-domain data), DIFT outperforms all the weakly-supervised baselines on all benchmarks by a large margin. It even outperforms the state-of-the-art supervised methods on PF-WILLOW, and for 9 out of 18 categories on SPair-71k.

**Qualitative Results.** To get a better understanding of DIFT's performance, we visualize a few correspondences on SPair-71k using various off-the-shelf features in Fig. 3. We observe that DIFT is able to identify correct correspondences under cluttered scenes, viewpoint changes, and instance-level appearance changes.

In addition to visualizing correspondence within the same categories in SPair-71k, we also visualize the correspondence established using $DIFT_{sd}$ across various categories in Fig. 4. Specifically, we select an image patch from a random image and query the image patches with the nearest DIFT embedding in the rest of the test split but from different categories. DIFT is able to identify correct correspondence across various categories.

**Sensitivity to the choice of time step $t$.** For $DIFT_{sd}$, we plot how its PCK `per point` varies with different choices of $t$ on SPair-71k in Fig. 5. DIFT is robust to the choice of $t$ on semantic correspondence, as a wide range of $t$ outperforms other off-the-shelf self-supervised features. Appendix B includes more discussion on how and why does $t$ affect the nature of correspondence.

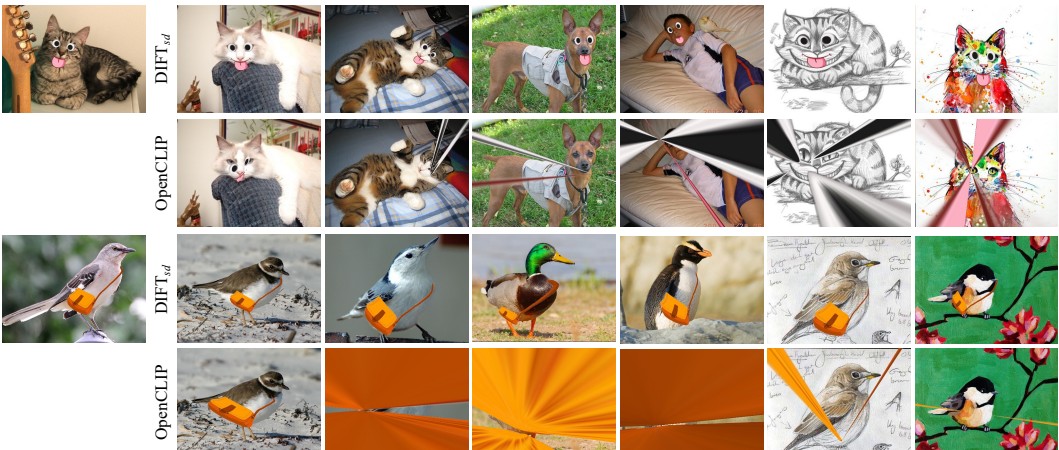

Figure 6: Edit propagation. The first column shows the source image with edits (i.e., the overlaid stickers), and the rest columns are the propagated results on new images from different instances, categories, and domains, respectively. Compared to OpenCLIP, DIFT$_{sd}$ propagates edits much more accurately. More results are in Fig. 20 of Appendix E.

## 5.3 Application: Edit Propagation

One application of DIFT is image editing: we can propagate edits in one image to others that share semantic correspondences. This capability is demonstrated in Fig. 6, where we showcase DIFT's ability to reliably propagate edits across different instances, categories, and domains, without any correspondence supervision.

To achieve this propagation, we simply compute a homography transformation between the source and target images using only matches found in the regions of the intended edits. By applying this transformation to the source image edits (e.g., an overlaid sticker), we can integrate them into the corresponding regions of the target image. Figure 6 shows the results for both OpenCLIP and DIFT$_{sd}$ using the same propagation techniques. OpenCLIP fails to compute reasonable transformation due to the lack of reliable correspondences. In contrast, DIFT$_{sd}$ achieves much better results, further justifying the effectiveness of DIFT in finding semantic correspondences.

# 6 Other Correspondence Tasks

We also evaluate DIFT on geometric correspondence and temporal correspondence. As in Sec. 5, we compare DIFT to other off-the-shelf self-supervised features as well as task-specific methods.

## 6.1 Geometric Correspondence

Intuitively, we find when $t$ is small, DIFT focuses more on low-level details, which makes it useful as a geometric feature descriptor.

**Setup.** We evaluate DIFT for homography estimation on the HPatches benchmark [4]. It contains 116 sequences, where 57 sequences have illumination changes and 59 have viewpoint changes. Following [91], we extract a maximum of 1,000 keypoints from each image, and use `cv2.findHomography()` to estimate the homography from mutual nearest neighbor matches. For DIFT, we use the same set of keypoints detected by SuperPoint [16], as in CAPS [91].

For evaluation, we follow the corner correctness metric used in [91]: the four corners of one image are transformed into the other image using the estimated homography and are then compared with those computed using the ground-truth homography. We deem the estimation correct if the average error of the four corners is less than $\epsilon$ pixels. Note that we do this evaluation on the original image resolution following [91], unlike [16].

**Results.** We report the accuracy comparison between DIFT and other methods in Tab. 4. Visualization of the matched points can be found in Fig. 7. Though not trained using any explicit geometry supervision, DIFT is still on par with the methods that utilize explicit geometric supervision signals

Table 4: Homography estimation accuracy [%] at 1, 3, 5 pixels on HPatches. Colors of numbers indicate the **best**, second-best results. All the DIFT results have `gray background` for better reference. DIFT with SuperPoint keypoints achieves competitive performance.

| Method | Geometric Supervision | All $\epsilon=1$ | $\epsilon=3$ | $\epsilon=5$ | Viewpoint Change $\epsilon=1$ | $\epsilon=3$ | $\epsilon=5$ | Illumination Change $\epsilon=1$ | $\epsilon=3$ | $\epsilon=5$ |
|---|---|---|---|---|---|---|---|---|---|---|
| SIFT [51] | None | 40.2 | 68.0 | 79.3 | 26.8 | 55.4 | 72.1 | 54.6 | 81.5 | 86.9 |
| LF-Net [59] | | 34.4 | 62.2 | 73.7 | 16.8 | 43.9 | 60.7 | 53.5 | 81.9 | 87.7 |
| SuperPoint [16] | | 36.4 | 72.7 | 82.6 | 22.1 | 56.1 | 68.2 | 51.9 | 90.8 | **98.1** |
| D2-Net [19] | Strong | 16.7 | 61.0 | 75.9 | 3.7 | 38.0 | 56.6 | 30.2 | 84.9 | 95.8 |
| DISK [86] | | 40.2 | 70.6 | 81.5 | 23.2 | 51.4 | 67.9 | 58.5 | 91.2 | 96.2 |
| ContextDesc [52] | | 40.9 | 73.0 | 82.2 | 29.6 | 60.7 | 72.5 | 53.1 | 86.2 | 92.7 |
| R2D2 [66] | | 40.0 | 74.4 | 84.3 | 26.4 | 60.4 | 73.9 | 54.6 | 89.6 | 95.4 |
| *w/ SuperPoint kp.* | | | | | | | | | | |
| CAPS [91] | Weak | 44.8 | **76.3** | **85.2** | **35.7** | **62.9** | **74.3** | 54.6 | 90.8 | 96.9 |
| DINO [10] | | 38.9 | 70.0 | 81.7 | 21.4 | 50.7 | 67.1 | 57.7 | 90.8 | 97.3 |
| DIFT$_{adm}$ (ours) | None | 43.7 | 73.1 | 84.8 | 26.4 | 57.5 | **74.3** | **62.3** | 90.0 | 96.2 |
| OpenCLIP [36] | | 33.3 | 67.2 | 78.0 | 18.6 | 45.0 | 59.6 | 49.2 | 91.2 | 97.7 |
| DIFT$_{sd}$ (ours) | | **45.6** | 73.9 | 83.1 | 30.4 | 56.8 | 69.3 | 61.9 | **92.3** | **98.1** |

Viewpoint Change         Illumination Change

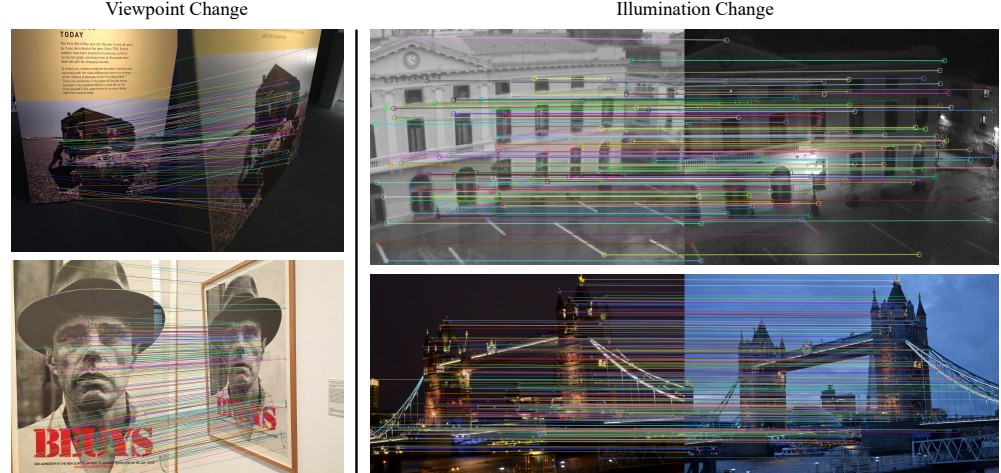

Figure 7: Sparse feature matching using DIFT$_{sd}$ on HPatches after removing outliers with `cv2.findHomography()`. Left are image pairs under viewpoint change, and right are ones under illumination change. Although never trained with correspondence labels, it works well under both challenging changes. More results are in Fig. 21 of Appendix E.

designed specifically for this task, such as correspondences obtained from Structure-from-Motion [73] pipelines. This shows that not only semantic-level correspondence, but also geometric correspondence emerges from image diffusion models.

## 6.2 Temporal Correspondence

DIFT also demonstrates strong performance on temporal correspondence tasks, including video object segmentation and pose tracking, although never trained or fine-tuned on such video data.

**Setup.** We evaluate DIFT on two challenging video label propagation tasks: (1) DAVIS-2017 video instance segmentation benchmark [65]; (2) JHMDB keypoint estimation benchmark [40].

Following evaluation setups in [49, 37, 10, 95], representations are used as a similarity function: we segment scenes with nearest neighbors between consecutive video frames. Note that there is no training involved in this label propagation process. We report region-based similarity $\mathcal{J}$ and contour-based accuracy $\mathcal{F}$ [64] for DAVIS, and PCK for JHMDB.

**Results.** Table 5 reports the experimental results, comparing DIFT with other self-supervised features (pre-)trained with or without video data. DIFT$_{adm}$ outperforms all the other self-supervised learning methods on both benchmarks, even surpassing models specifically trained on video data by a significant margin. DIFT also yields the best results within the same pre-training dataset.

Table 5: Video label propagation results on DAVIS-2017 and JHMDB. Colors of numbers indicate the **best**, second-best results. All the DIFT results have gray background for better reference. DIFT even outperforms other self-supervised learning methods specifically trained with video data.

| (pre-)Trained on Videos | Method | Dataset | DAVIS | | | JHMDB | |
|---|---|---|---|---|---|---|---|
| | | | $\mathcal{J}\&\mathcal{F}_m$ | $\mathcal{J}_m$ | $\mathcal{F}_m$ | PCK@0.1 | PCK@0.2 |
| ✗ | InstDis [93] | | 66.4 | 63.9 | 68.9 | 58.5 | 80.2 |
| | MoCo [30] | | 65.9 | 63.4 | 68.4 | 59.4 | 80.9 |
| | SimCLR [12] | ImageNet [15] | 66.9 | 64.4 | 69.4 | 59.0 | 80.8 |
| | BYOL [25] | w/o labels | 66.5 | 64.0 | 69.0 | 58.8 | 80.9 |
| | SimSiam [13] | | 67.2 | 64.8 | 68.8 | 59.9 | 81.6 |
| | DINO [10] | | 71.4 | 67.9 | 74.9 | 57.2 | 81.2 |
| | DIFT$_{adm}$ (ours) | | 75.7 | 72.7 | 78.6 | 63.4 | 84.3 |
| | OpenCLIP [36] | LAION [75] | 62.5 | 60.6 | 64.4 | 41.7 | 71.7 |
| | DIFT$_{sd}$ (ours) | | 70.0 | 67.4 | 72.5 | 61.1 | 81.8 |
| ✓ | VINCE [24] | | 65.2 | 62.5 | 67.8 | 58.8 | 80.4 |
| | VFS [95] | | 68.9 | 66.5 | 71.3 | 60.9 | 80.7 |
| | UVC [49] | Kinetic [11] | 60.9 | 59.3 | 62.7 | 58.6 | 79.6 |
| | CRW [37] | | 67.6 | 64.8 | 70.2 | 58.8 | 80.3 |
| | Colorization [88] | | 34.0 | 34.6 | 32.7 | 45.2 | 69.6 |
| | CorrFlow [44] | OxUvA [87] | 50.3 | 48.4 | 52.2 | 58.5 | 78.8 |
| | TimeCycle [92] | VLOG [21] | 48.7 | 46.4 | 50.0 | 57.3 | 78.1 |
| | MAST [43] | YT-VOS [96] | 65.5 | 63.3 | 67.6 | - | - |
| | SFC [34] | | 71.2 | 68.3 | 74.0 | 61.9 | 83.0 |

Figure 8: Video label propagation results on DAVIS-2017. Colors indicate segmentation masks for different instances. Blue rectangles show the first frames. Compared to DINO, DIFT$_{adm}$ produces masks with more accurate and sharper boundaries. More results are in Fig. 22 of Appendix E.

We also show qualitative results in Fig. 8, presenting predictions of video instance segmentation in DAVIS, comparing DIFT$_{adm}$ with DINO. DIFT$_{adm}$ produces masks with clearer boundaries when single or multiple objects are presented in the scene, even attends well to objects in the presence of occlusion.

## 7    Conclusion

This paper demonstrates that correspondence emerges from image diffusion models without explicit supervision. We propose a simple technique to extract this implicit knowledge as a feature extractor named DIFT. Through extensive experiments, we show that although without any explicit supervision, DIFT outperforms both weakly-supervised methods and other off-the-shelf self-supervised features in identifying semantic, geometric and temporal correspondences, and even remains on par with the state-of-the-art supervised methods on semantic correspondence. We hope our work inspires future research on how to better utilize these emergent correspondence from image diffusion, as well as rethinking diffusion models as self-supervised learners.

**Acknowledgement.** This work was partially funded by NSF 2144117 and the DARPA Learning with Less Labels program (HR001118S0044). We would like to thank Zeya Peng for her help on the edit propagation section and the project page, thank Kamal Gupta for sharing the evaluation details in the ASIC paper, thank Aaron Gokaslan, Utkarsh Mall, Jonathan Moon, Boyang Deng, and all the anonymous reviewers for valuable discussion and feedback.

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

# A  Societal Impact

Although DIFT can be used with any diffusion model parameterized with a U-Net, the dominant publicly available model is the one trained on LAION [75]. The LAION dataset has been identified as having several issues including racial bias and stereotypes [7]. Diffusion models trained on these datasets inherit these issues. While these issues may a priori seem less important for estimating correspondences, it might lead to differing accuracies for different kinds of images. One could obtain the benefit of good correspondences without the associated issues if one could trained a diffusion model on a curated dataset. Unfortunately, the huge computational cost also prohibits the training of diffusion models in academic settings on cleaner datasets. We hope that our results encourage efforts to build more carefully trained diffusion models.

# B  Discussion

**Why does correspondence emerge from image diffusion?** One conjecture is that the diffusion training objective (i.e., coarse-to-fine reconstruction loss) requires the model to produce good, informative features for every pixel. This is in contrast to DINO and OpenCLIP that use image-level contrastive learning objectives. In our experiments, we have attempted to evaluate the importance of the training objective by specifically comparing $\text{DIFT}_{adm}$ and DINO in all our evaluations: two models that share exactly the same training data, i.e., ImageNet-1k without labels.

**How and why does $t$ affect the nature of correspondence?** In Fig. 9, for the same clean image, we first add different amount of noise to get different $x_t$ following Eq. (2), then feed it into SD's denoising network $\epsilon_\theta$ together with time step $t$ to get the predicted clean image $\hat{x}_0^t = \frac{x_t - (\sqrt{1-\alpha_t})\epsilon_\theta(x_t, t)}{\sqrt{\alpha_t}}$. We can see that, with the increase of $t$, more and more details are removed and only semantic-level features are preserved, and when $t$ becomes too large, even the object structure is distorted. Intuitively, this explains why we need a small $t$ for correspondences that requires details and a relatively large $t$ for semantic correspondence.

input clean image          predicted clean images at different time step $t$

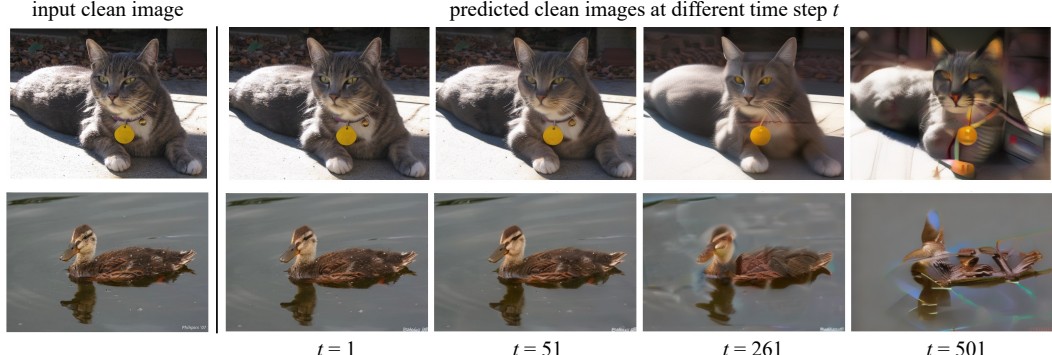

$t = 1$      $t = 51$      $t = 261$      $t = 501$

Figure 9: Within a reasonable range, when $t$ gets larger, the predicted clean images remain the overall structure but have less details, suggesting DIFT contains more semantic-level information and less low-level features with the increase of $t$.

**How long does it take to run DIFT?** Since we only perform a single inference step when extracting DIFT, it actually takes similar running time compared to competing self-supervised features with the same input image size. For example, when extracting features for semantic correspondence as in Sec. 5, on one single NVIDIA A6000 GPU, $\text{DIFT}_{sd}$ takes 203 ms vs. OpenCLIP's 231 ms on one single 768×768 image; $\text{DIFT}_{adm}$ takes 110 ms vs. DINO's 154 ms on one single 512×512 image. In practice, as mentioned in the last paragraph of Sec. 4.2, since there is randomness when extracting DIFT, we actually use a batch of random noise to get an averaged feature map for each image to slightly boost stability and performance, which would increase the running time shown above. But if computation is a bottleneck, one can remove this optimization at the cost of a tiny loss in performance: e.g., on SPair-71k, $\text{DIFT}_{sd}$: PCK 59.5→57.9; $\text{DIFT}_{adm}$: PCK 52.0→51.1.

**Would diffusion inversion help?** Another way to get $x_t$ from a real input image is diffusion inversion. Using DDIM inversion [77] to recover input image's corresponding $x_t$ and then feeding into $f_\theta$ to

get diffusion feature yielded similar results. At the same time, inversion makes the inference process several times slower. We leave how to utilize diffusion inversion to get better correspondence to future work.

**Does correspondence information exist in SD's encoder?** We also evaluated SD's VAE encoder's performance on all benchmarks and found that its performance was lower by an order of magnitude. So DIFT$_{sd}$'s correspondence only emerges inside its U-Net and requires diffusion-based training.

**Would task-specific adaptation lead DIFT to better results?** More sophisticated mechanisms could be applied to further enhance the diffusion features, e.g., concatenating and re-weighting features from different time step $t$ and different network layers, or even fine-tuning the network with task-specific supervision. Some recent works [5, 94, 101] fine-tune either the U-Net or the attached head for dense prediction tasks and yield better performance. However, task-specific adaptation entangles the quality of the features themselves with the efficacy of the fine-tuning procedure. To keep the focus on the representation, we chose to avoid any fine-tuning to demonstrate the quality of the off-the-shelf DIFT. Nevertheless, our preliminary experiments suggest that such fine-tuning would indeed further improve performance on correspondence. We'll leave how to better adapt DIFT to downstream tasks to future work.

## C   Implementation Details

The total time step $T$ for both diffusion models (ADM and SD) is 1000. U-Net consists of downsampling blocks, middle blocks and upsampling blocks. We only extract features from the upsampling blocks. ADM's U-Net has 18 upsampling blocks and SD's U-Net has 4 upsampling blocks (the definition of blocks are different between these two models). Feature maps from the $n$-th upsampling block output are used as the final diffusion feature. For a fair comparison, we also grid-search which layer to extract feature for DINO and OpenCLIP for each task, and report the best results among the choices.

As mentioned in the last paragraph of Sec. 4.2, when extracting features for one single image using DIFT, we use a batch of random noise to get an averaged feature map. The batch size is 8 by default. We shrink it to 4 when encountering GPU memory constraints.

The input image resolution varies across different tasks but we always keep it the same within the comparison vs. other off-the-shelf self-supervised features (i.e., DIFT$_{adm}$ vs. DINO, DIFT$_{sd}$ vs. OpenCLIP) thus the comparisons are fair. For DIFT, feature map size and dimension also depend on which U-Net layer features are extracted from.

The following sections list the time step $t$ and upsampling block index $n$ ($n$ starts from 0) we used for each DIFT variant on different tasks as well as input image resolution and output feature map tensor shape.

### C.1   Semantic Correspondence

We use $t = 101$ and $n = 4$ for DIFT$_{adm}$ on input image resolution $512 \times 512$ so feature map size is $1/16$ of input and dimension is 1024; we use $t = 261$ and $n = 1$ for DIFT$_{sd}$ on input image resolution $768 \times 768$ so feature map size is $1/16$ of input and dimension is 1280. These hyper-parameters are shared on all semantic correspondence tasks including SPair-71k, PF-WILLOW, and CUB, as well as the visualizations in Figs. 1, 15 and 16.

We don't use image-specific prompts for DIFT$_{sd}$. Instead, we use a general prompt "a photo of a [class]" where [class] denotes the string of the input images' category, which is given by the dataset. For example, for the images of SPair-71k under cat class, the prompt would be "a photo of a cat". For CUB, the same prompt is used for all images: "a photo of a bird". Changing per-class prompt to a null prompt (empty string "") will only lead a very small performance drop, e.g., DIFT$_{sd}$'s PCK per point on SPair-71k: 59.5→57.6.

## C.2  Geometric Correspondence

On HPatches, the input images are resized to $768\times768$ to extract features for both $\text{DIFT}_{adm}$ and $\text{DIFT}_{sd}$. We use $t = 41$, $n = 11$ for $\text{DIFT}_{adm}$ so feature map size is $1/2$ of input and dimension is 512; we use $t = 0$, $n = 2$ for $\text{DIFT}_{sd}$ so feature map size is $1/8$ of input and dimension is 640.

In addition, for $\text{DIFT}_{sd}$, each image's prompt is a null prompt, i.e., an empty string `""`.

For all the methods listed in Tab. 4, when doing homography estimation, we tried both cosine and L2 distance for mutual nearest neighbor matching, and both `RANSAC` and `LMEDS` for `cv2.findHomography()`, and eventually we report the best number among these choices for each method.

## C.3  Temporal Correspondence

The configurations we use for $\text{DIFT}_{adm}$ and $\text{DIFT}_{sd}$ are:

| Dataset | Method | Time step $t$ | Block index $n$ | Temperature for softmax | Propagation radius | $k$ for top-$k$ | Number of prev. frames |
|---|---|---|---|---|---|---|---|
| DAVIS-2017 | $\text{DIFT}_{adm}$ | 51 | 7 | 0.1 | 15 | 10 | 28 |
| DAVIS-2017 | $\text{DIFT}_{sd}$ | 51 | 2 | 0.2 | 15 | 15 | 28 |
| JHMDB | $\text{DIFT}_{adm}$ | 101 | 5 | 0.2 | 5 | 15 | 28 |
| JHMDB | $\text{DIFT}_{sd}$ | 51 | 2 | 0.1 | 5 | 15 | 14 |

For experiments on DAVIS, we use the same original video frame size (480p version of DAVIS, specific size varies across different videos) as in DINO's implementation [10], for both $\text{DIFT}_{adm}$ and $\text{DIFT}_{sd}$. n=7 for $\text{DIFT}_{adm}$ so feature map size is $1/8$ of input and dimension is 512. n=2 for $\text{DIFT}_{sd}$ so feature map size is $1/8$ of input and dimension is 640. For experiments on JHMDB, following CRW's implementation [37], we resize each video frame's smaller side to 320 and keep the original aspect ratio. n=5 for $\text{DIFT}_{adm}$ so feature map size is $1/8$ of input and dimension is 1024. n=2 for $\text{DIFT}_{sd}$ so feature map size is $1/8$ of input and dimension is 640.

In addition, for $\text{DIFT}_{sd}$, each image's prompt is a null prompt, i.e., an empty string `""`.

# D  Additional Quantitative Results

## D.1  Semantic Correspondence on PF-PASCAL

We didn't do evaluation on PF-PASCAL [27] in the main paper because we found over half of the test images (i.e., 302 out of 506) actually also appear in the training set, which makes the benchmark numbers much less convincing, and also partially explains why the previous supervised methods tend to have much higher test accuracy on PF-PASCAL vs. PF-WILLOW (e.g., over 90 vs. around 70) even using exactly the same trained model. And this duplication issue of train/test images also gives huge unfair disadvantage to the methods that are never adapted (either supervised or unsupervised) on the training set before evaluation.

However, even in this case, as shown in Tab. 6, DIFT still demonstrates competitive performance compared to the state-of-the-art weakly-supervised method PWarpC [83], as well as huge gains vs. other off-the-shelf self-supervised features.

Table 6: PCK `per image` on PF-PASCAL. The groupings and colors follow Tab. 1.

| Sup. | Method | PCK@$\alpha_{img}$ | |
|---|---|---|---|
| | | $\alpha = 0.05$ | $\alpha = 0.10$ |
| (b) | PWarpC [83] | 64.2 | 84.4 |
| (c) | DINO [10] | 36.9 | 53.6 |
| | $\text{DIFT}_{adm}$ (ours) | 56.5 | 72.5 |
| | OpenCLIP [36] | 39.8 | 61.1 |
| | $\text{DIFT}_{sd}$ (ours) | **69.4** | **84.6** |

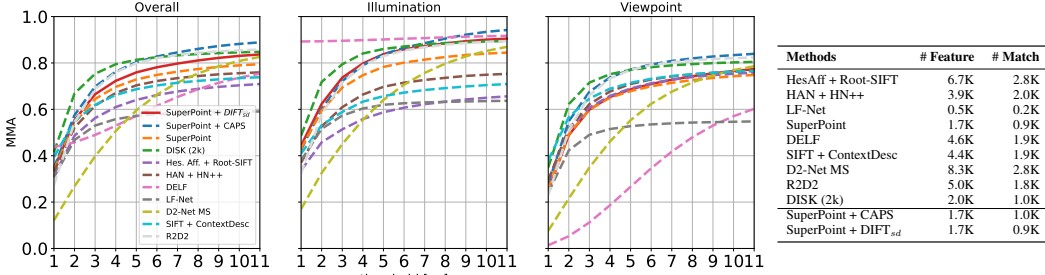

Figure 10: Mean Matching Accuracy (MMA) on HPatches [4]. For each method, we show the MMA with varying pixel error thresholds. We also report the mean number of detected features and mutual nearest neighbor matches. Although not trained with any explicit geometric correspondence labels, $\text{DIFT}_{sd}$ is able to achieves competitive performance compared to other feature descriptors that are specifically design or trained for this task.

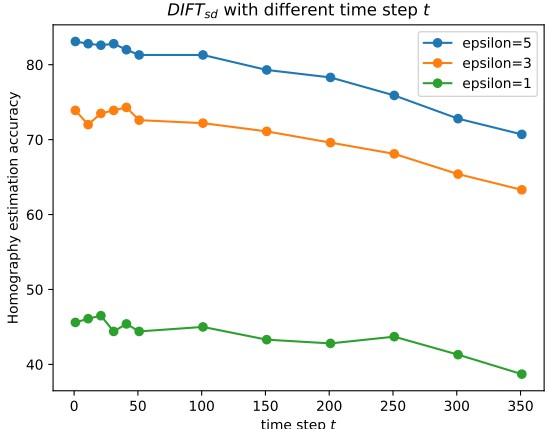

Figure 11: Homography estimation accuracy [%] at 1, 3, 5 pixels on HPatches using $\text{DIFT}_{sd}$ with different time step $t$. Intuitively, as $t$ gets larger, DIFT contains more semantic information and less low-level details, so the accuracy decreases when using larger $t$.

## D.2 Feature Matching on HPatches

Following CAPS [91], for the given image pair, we extract SuperPoint [16] keypoints in both images and match them using our proposed feature descriptor, $\text{DIFT}_{sd}$. We follow the evaluation protocol as in [19, 91] and use Mean Matching Accuracy (MMA) as the evaluation metric, where only mutual nearest neighbors are considered as matched points. The MMA score is defined as the average percentage of correct matches per image pair under a certain pixel error threshold. Fig. 10 shows the comparison between DIFT and other feature descriptors that are specially designed or trained for geometric correspondence. We report the average results for the whole dataset, as well as subsets on illumination and viewpoint changes respectively. For each method, we also present the mean number of detected features per image and mutual nearest neighbor matches per image pair. We can see that, although not trained with any explicit geometry supervision, DIFT is still able to achieve competitive performance.

## D.3 Analysis on Hyper-parameters

Here we analyze how the choice of time step and U-Net layer would affect DIFT's performance on different correspondence tasks.

**Ablation on time step.** Similar to Fig. 5, we plot how HPatches homography estimation accuracy and DAVIS video label propagation accuracy vary with different choices of $t$, in Figs. 11 and 12

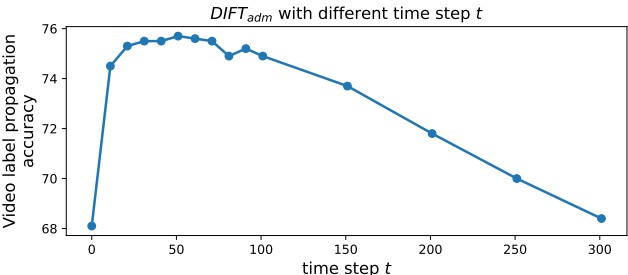

Figure 12: Video label propagation accuracy ($\mathcal{J}\&\mathcal{F}_m$) on DAVIS using $\text{DIFT}_{adm}$ with different time step $t$. There's a wide range of $t$, where DIFT maintains a stable and competitive performance.

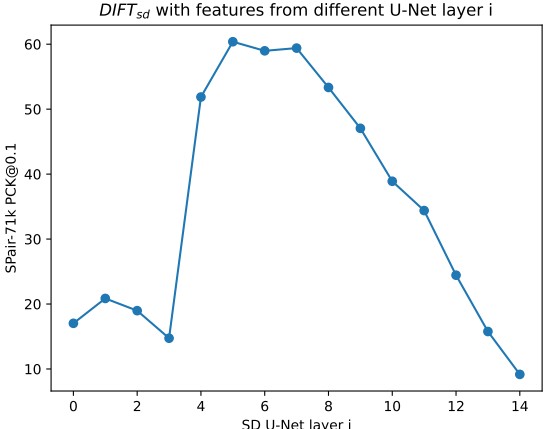

Figure 13: PCK `per point` on SPair-71k using $\text{DIFT}_{sd}$ with different layer $i$ inside U-Net's 15 upsampling layers in total. The transition from block index $n$ to layer index $i$ is 0/1/2/3 to 3/7/11/14 respectively. We can see there are multiple choices of $i$ leading to good performance.

respectively. Both curves have smooth transitions and there's a large range of $t$ where DIFT gives competitive performance.

**Ablation on U-Net layer.** Compared to the definition of 4 block choices in Appendix C, here we make a more fine-grained sweep over SD's 15 layers inside U-Net upsampling blocks. The transition from *block* index $n$ to *layer* index $i$ is 0/1/2/3 to 3/7/11/14 respectively and both start from 0. We evaluate PCK `per point` on SPair-71k using $\text{DIFT}_{sd}$ with different layer index $i$. As shown in Fig. 13, the accuracy varies but there are still multiple choices of $i$ that lead to good performance.

# E   Additional Qualitative Results

**PCA visualization of DIFT.** In Fig. 14, for each pair of images, we extract $\text{DIFT}_{sd}$ from the segmented instances, then compute PCA and visualize the first 3 components, where each component serves as a color channel. We can see the same object parts share similar embeddings, which also demonstrates the emergent correspondence.

**Correspondence on diverse internet images.** Same as Fig. 1, in Figs. 15 and 16 we show more correspondence prediction on various image groups that share similar semantics. For each target image, the $\text{DIFT}_{sd}$ predicted point will be displayed as a red circle, together with a heatmap showing the per-pixel cosine distance calculated using $\text{DIFT}_{sd}$. We can see it works well across instances, categories, and even image domains, e.g., from an umbrella photo to an umbrella logo.

**Semantic correspondence comparison among off-the-shelf features on SPair-71k.** Same as Fig. 3, we show more comparison in Fig. 17, where we can see DIFT works well under challenging occlusion, viewpoint change and intra-class appearance variation.

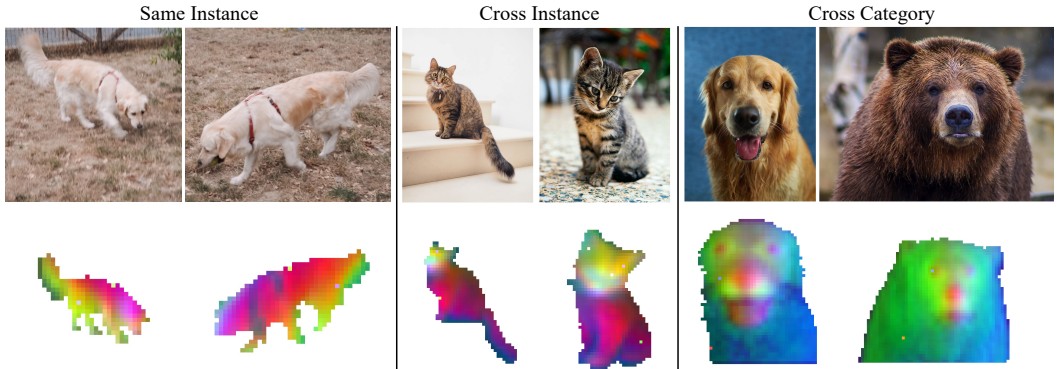

Figure 14: Visualization of the first three PCA components of $DIFT_{sd}$ on the segmented instance pairs (same instance, cross instance, cross category). Each component matches a color channel. We can see the same object parts share similar DIFT embeddings.

**Cross-category semantic correspondence.** Same as Fig. 4, in Fig. 18 we select an interesting image patch from a random source image and query the image patches with the nearest $DIFT_{sd}$ features in the rest of the test split but with different categories. We see that DIFT is able to identify reasonable correspondence across various categories.

**Failure Cases on SPair-71k.** In Fig. 19, we select four examples with low PCK accuracy and visualize $DIFT_{sd}$'s predictions along with ground-truths. We can see that, when the semantic definition of key points is ambiguous, or the appearance/viewpoint change between source and target images is too dramatic, DIFT fails to give correct predictions.

**Image editing propagation.** Similar to Fig. 6, Fig. 20 shows more examples on edit propagation using our proposed $DIFT_{sd}$. It further demonstrates the effectiveness of DIFT on finding semantic correspondence, even when source image and target image are from different categories or domains.

**Geometric correspondence.** Same as Fig. 7, in Fig. 21 we show the sparse feature matching results using $DIFT_{sd}$ on HPatches. Though not trained using any explicit geometry supervision, DIFT still works well under large viewpoint change and challenging illumination change.

**Temporal correspondence.** Similar to Fig. 8, Fig. 22 presents additional examples of video instance segmentation results on DAVIS-2017, comparing DINO, $DIFT_{adm}$ and Ground-truth (GT). We can see $DIFT_{adm}$ could create instance masks that closely follow the silhouette of instances.

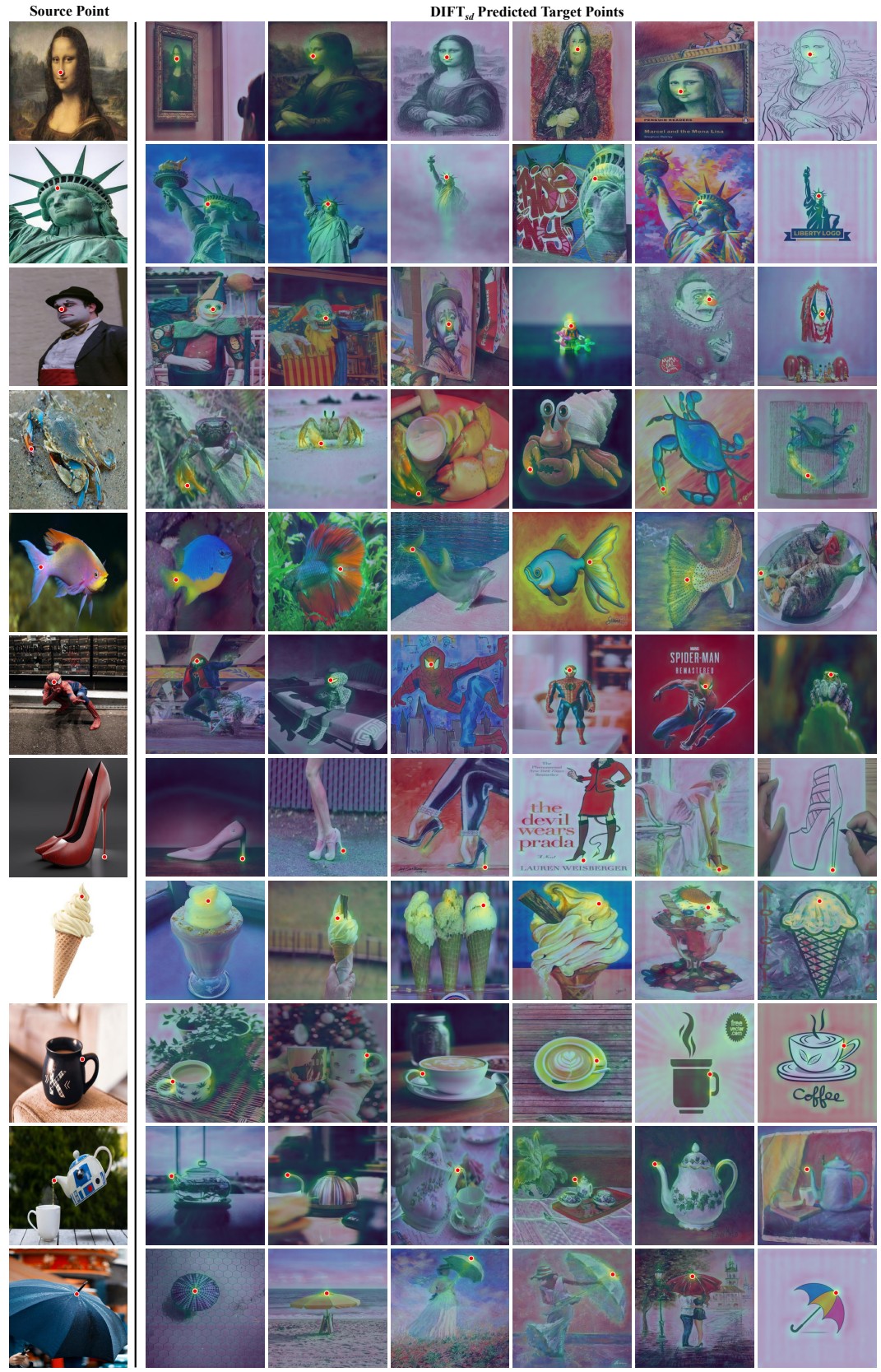

Figure 15: DIFT can find correspondences on real images across instances, categories, and even domains, e.g., from a photo of statue of liberty to a logo.

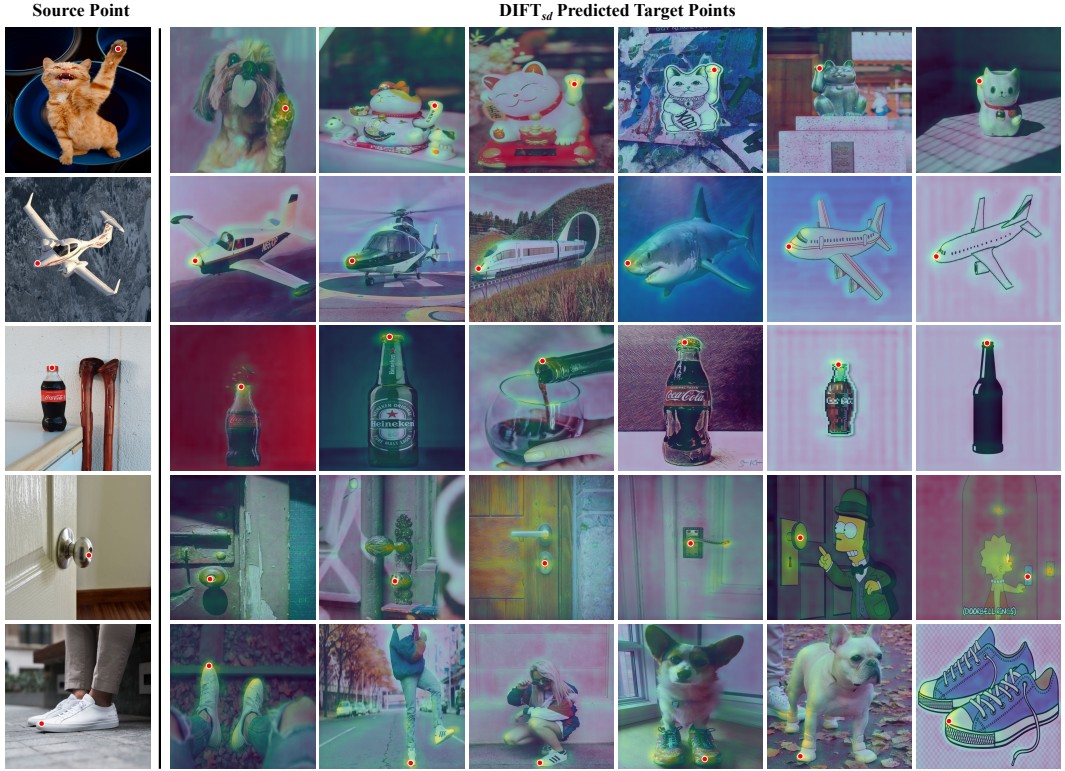

Figure 16: DIFT can find correspondences on real images across instances, categories, and even domains, e.g., from a photo of an aeroplane to a sketch.

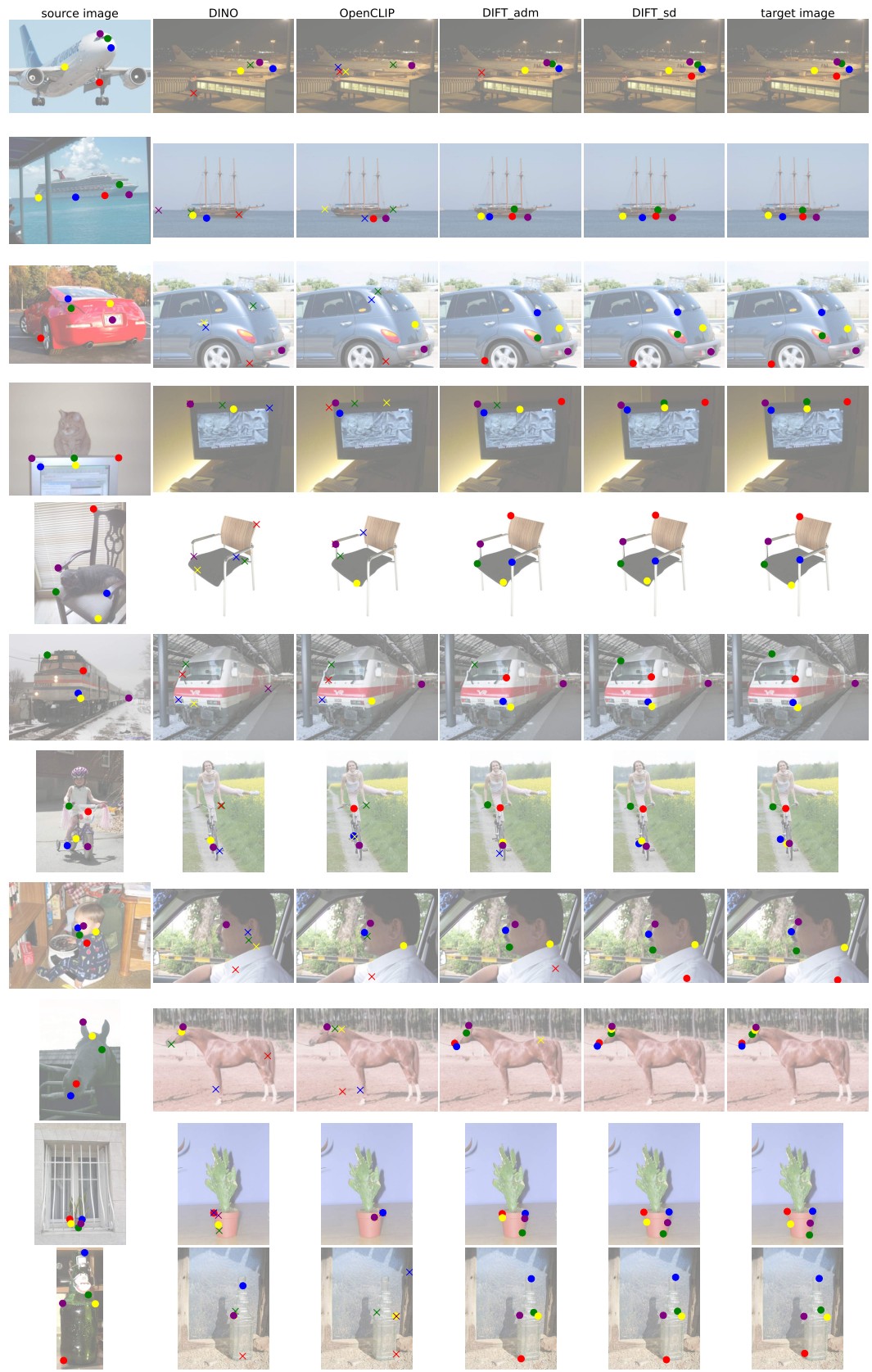

Figure 17: Semantic correspondence using various off-the-shelf features on SPair-71k. Circles indicates correct predictions while crosses for incorrect ones.

Source patch      Top-5 nearest neighbor cross-category target patches predicted by DIFT$_{sd}$

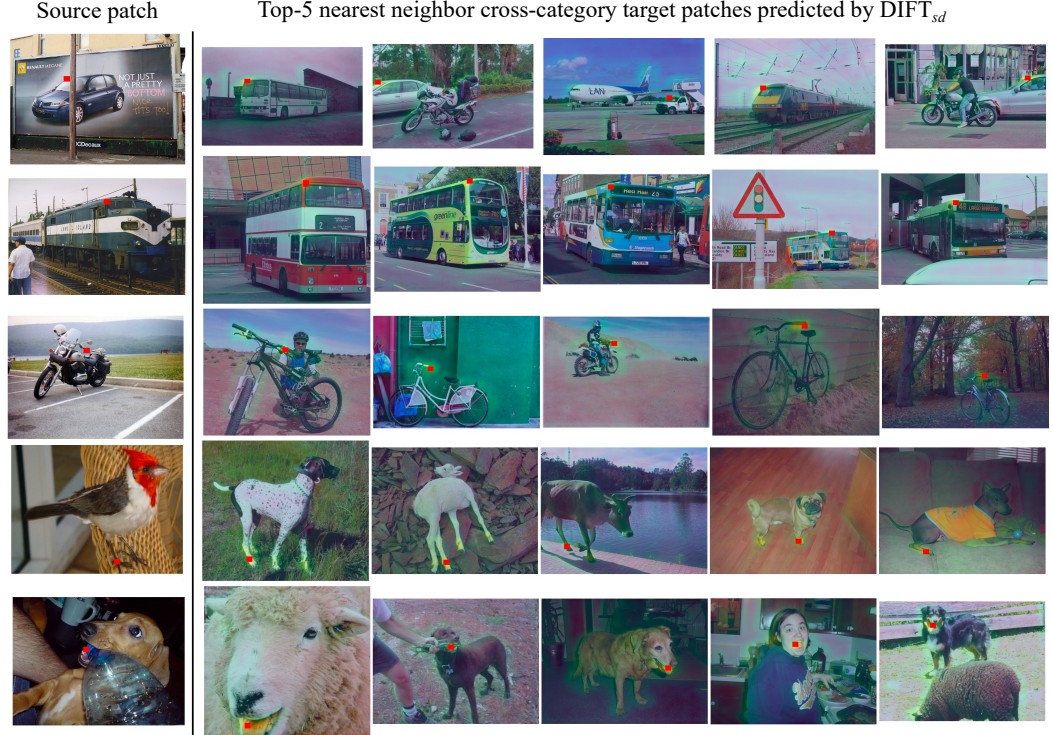

Figure 18: Given the image patch specified in the leftmost image (red dot), we use DIFT$_{sd}$ to query the top-5 nearest image patches from different categories in the SPair-71k test set. DIFT is still able to find correct correspondence for object parts with different overall appearance but sharing the same semantic meaning, e.g., the leg of a bird vs. the leg of a dog.

source image    DIFT$_{sd}$    target image      source image    DIFT$_{sd}$    target image

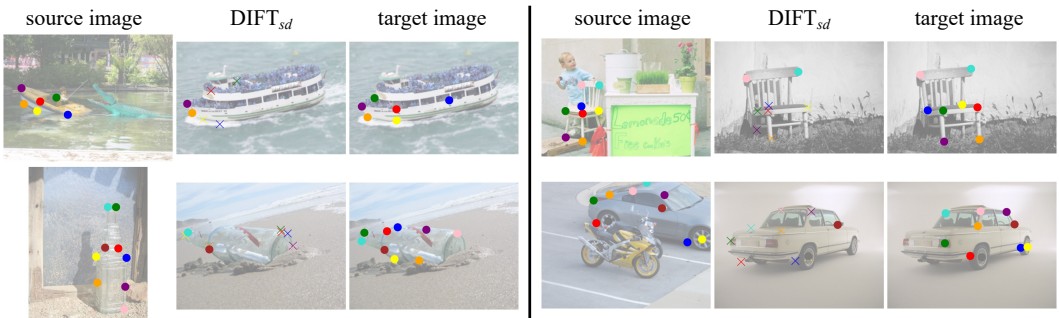

Figure 19: Failure cases of semantic correspondence on SPair-71k. Circle denotes correct predictions while cross for wrong ones. When the semantic definition of key points is ambiguous, or the appearance/viewpoint change between source and target images is too dramatic, DIFT$_{sd}$ fails to predict the correct corresponding points.

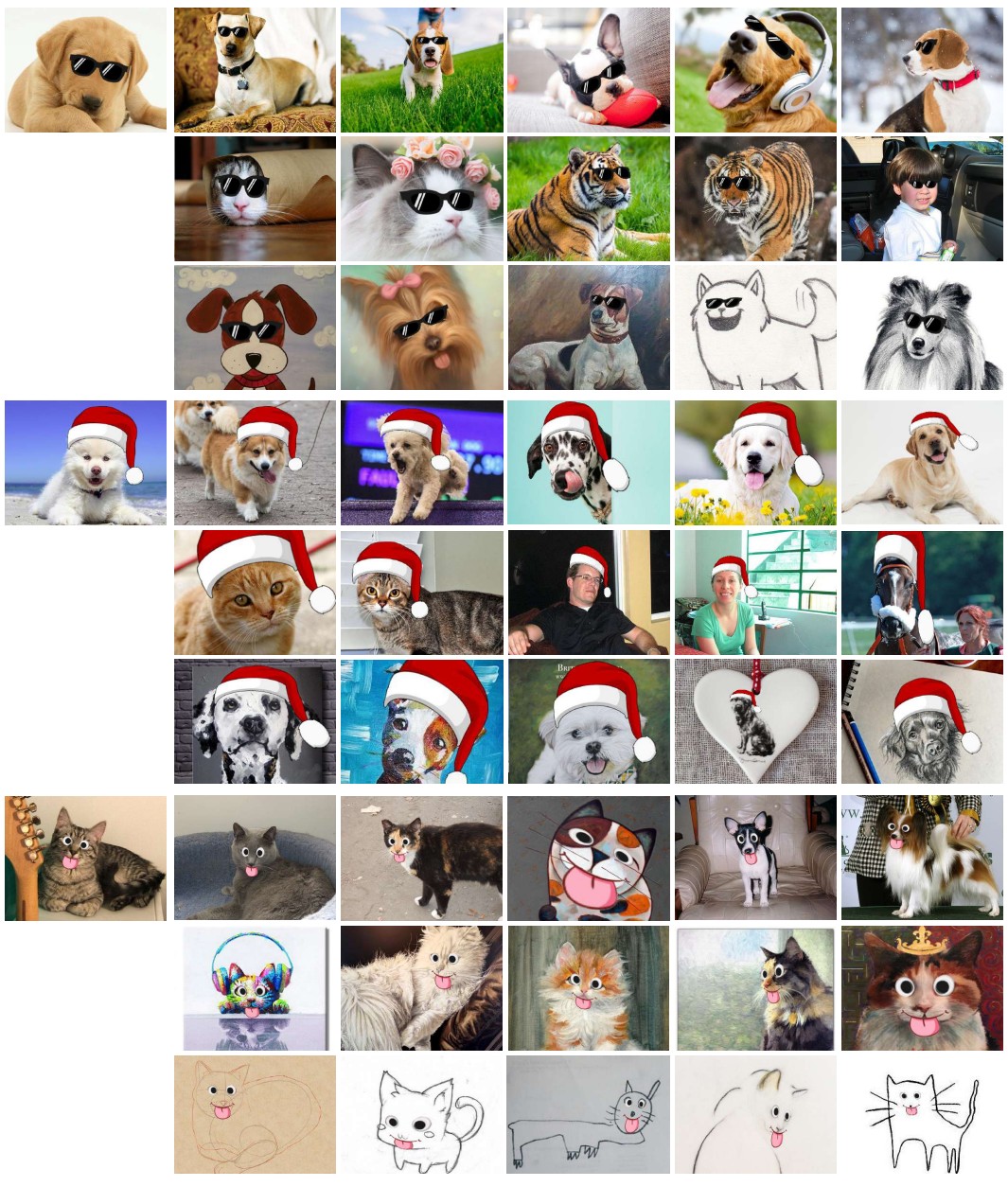

Figure 20: Edit propagation using DIFT$_{sd}$. Far left column: edited source images. Right columns: target images with the propagated edits. Note that despite the large domain gap in the last row, DIFT$_{sd}$ still manages to establish reliable correspondences for correct propagation.

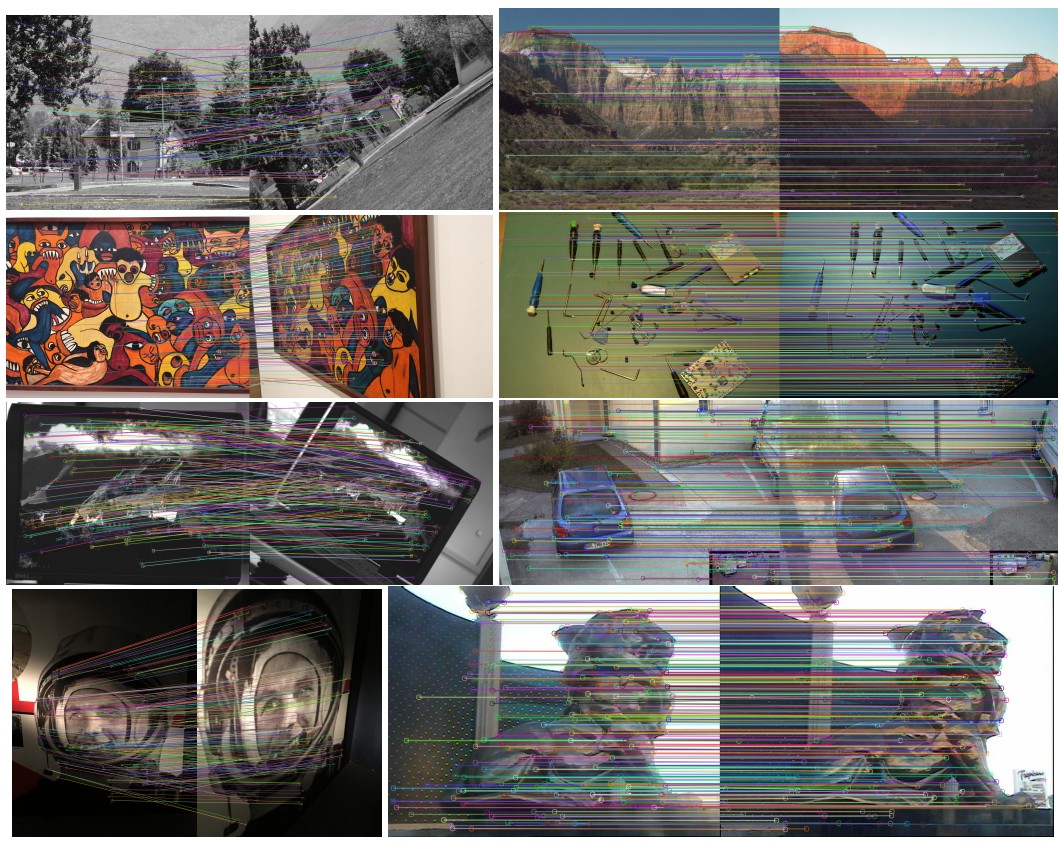

Figure 21: Sparse feature matching using DIFT$_{sd}$ on HPatches after removing outliers. Left are image pairs under viewpoint change, and right are ones under illumination change. Although never trained with geometric correspondence labels, it works well under both challenging changes.

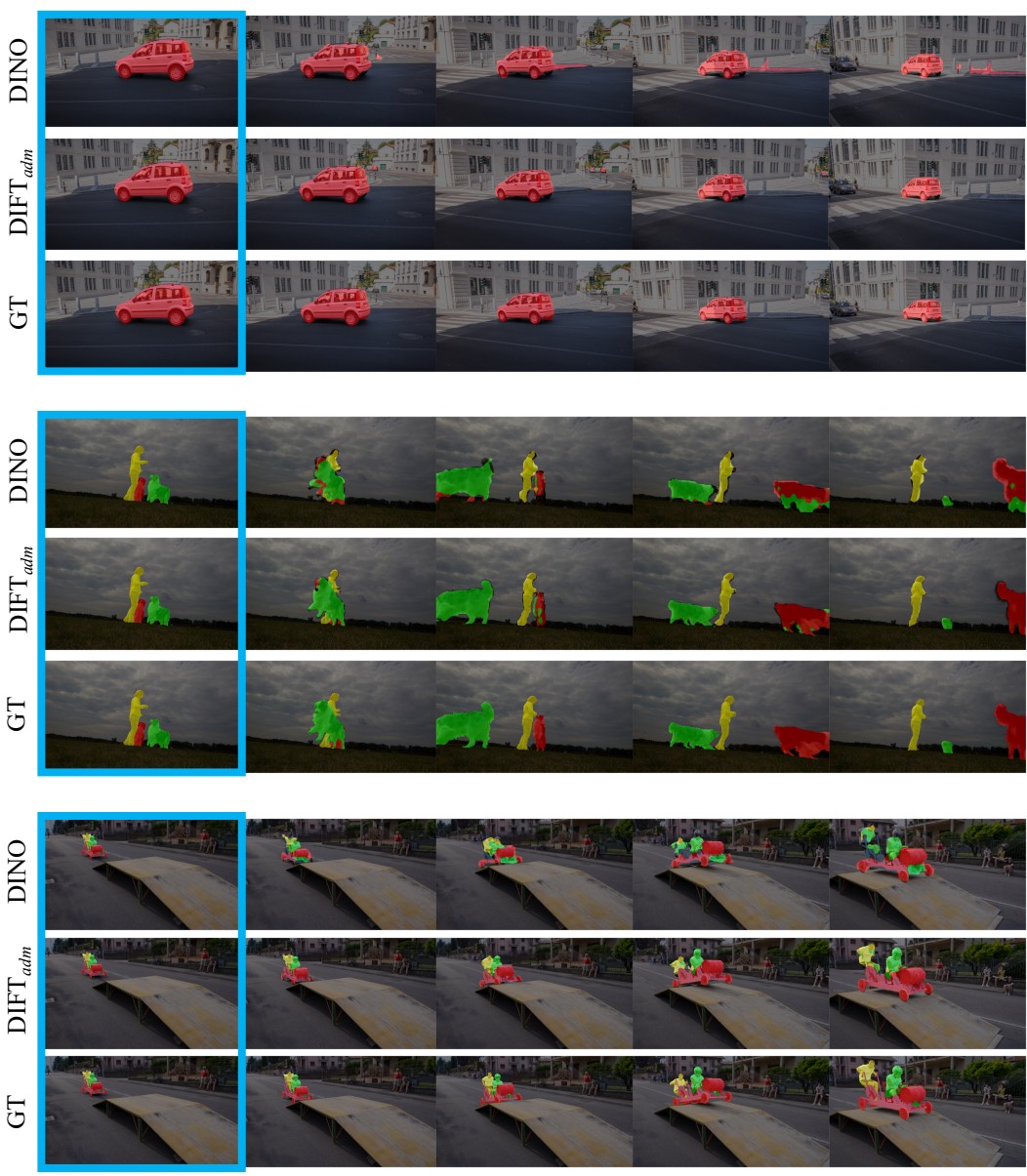

Figure 22: Additional video label propagation results on DAVIS-2017. Colors indicate segmentation masks for different instances. Blue rectangles show the first frames. GT is short for "Ground-Truth".

