# OpenReview forum: "Emergent Correspondence from Image Diffusion"
_NeurIPS.cc/2023/Conference — NeurIPS 2023 poster_

### Official Review · Reviewer_rXbs · 2023-06-13

**Soundness:** 3 good
**Presentation:** 3 good
**Contribution:** 3 good
**Rating:** 7
**Confidence:** 4

**Summary:**

The authors find that intra and cross-category correspondences are implicitly learnt by diffusion models trained self-supervised on large datasets. The paper proposes an approach to extract this knowledge as features from pre-trained Unet-based diffusion models. In particular, to compute the features of a particular image, noise is added to the image to simulate the diffusion process, and then input to the pre-trained diffusion model. The intermediate layer activations from the Unet at a particular timestep are used as features. They showcase the performance of these features on semantic-matching, outperforming other unsupervised baselines and strongly-supervised approaches specifically designed for semantic matching. They also show that without any task specific finetuning, the features can be used for geometric matching and temporal matching ( video-object-segmentation throught label propagation), with competitive performance compared to state-of-the-art.

**Strengths:**

- The proposed idea is simple and effective

- The results are convincing. Having a single model applicable to many different correspondence tasks without specific architecture/training and outperform task-specific methods would be a useful contribution. The new perspectives that it opens are exciting.

- The paper overall reads well.


**Weaknesses:**

1) The authors do not comment on the run-time of the proposed approach. I expect it will be quite slow since the image needs to go through the reverse diffusion process.

2) Related to the above, relying on the diffusion model at inference time makes the method impractical in many applications. To remove the reliance on diffusion models at inference, would it be possible to train a feature predictor, using as ground-truth the features extracted from a pre-trained models? Have you done any experiments in this direction?


**Questions:**

A) What is the run-time of the approach to extract feature on a single image?

B) I think details on the ‘architecture’ of the extracted features are missing. It is only said L.113-114 that “intermediate layer activations” are extracted from the unet, and that each depends on the task. I would appreciate to have some details there - what layers are extracted (for the different tasks), what resolution and channel dimension do they have, are they interpolated? Are features from multiple levels used? If so how are they aggregated?

C) What is the resolution of the extracted feature and at what resolution is the matching done? Is a similar resolution used in competitor works like DINO?
The resolution of the feature maps is of crucial importance for obtaining fine-grained/accurate correspondences. Since matching is done by computing all-to-all similarities between the two images, It has often limited previous approaches  to small resolution because of memory constraint. Just using a higher resolution feature map would lead to a significant improvement in results, not necessarily related to the features themselves.

D) Some misclassification and missing comparisons in semantic matching:

D.1) CNNGeo and A2Net are unsupervised - they only use single images as supervision, as opposed to NCNet for example which requires pairs of similar images

D.2) Comparison to recent approches are missing. In particular PWarpC [Truong et al. CVPR 2022] obtained state-of-the-art results in weakly-supervised semantic matching (with extension to supervised)

E) Would it be possible to extract features from multiple layers of the diffusion model such as to build a feature pyramid (similar to VGG or ResNet)? This could potentially be used as a plug-in replacement for backbones in correspondence approaches with a decoder.


F) How many different passes through the network (ie different random noise) are used for an image?



**Limitations:**

-

---

> ### Author Rebuttal · Authors · 2023-08-06
>
> Thank you for the insightful feedback! Please see our response below:
>
> **Run-time of DIFT**.
>
> Please refer to point 2 of global rebuttal above.
>
> Briefly speaking, DIFT is actually fast because it doesn’t need to run diffusion inversion thus only one network inference is involved (see the second last paragraph of Sec. 4.2). Also, since we only need the intermediate U-Net features, the inference process can stop immediately when reaching the desired block and doesn't need to run the following layers. For example, it only takes 203 ms for DIFT$\_{sd}$ vs. 231 ms for OpenCLIP on one single image. In addition, we also discuss the usage of diffusion inversion in the first paragraph of Appendix B.
>
> **Train a feature predictor network to have faster diffusion features**.
>
> Given the speed of DIFT is already pretty fast as mentioned above, we haven’t explored the direction to train a feature predictor network such that it can make DIFT even faster. But we believe the techniques in the field of knowledge distillation (e.g., train a smaller student network to distill knowledge from a much larger teacher network) could be generally applied to diffusion features, especially when latency of DIFT becomes a main bottleneck in practical usage.
>
> **Implementation details of DIFT including hyperparameters and resolutions**.
>
> We include the implementation details in Appendix C including the time step $t$ and network block index $n$ used in each task. We’ll add a pointer in the main paper in the future version. For input image size, feature map size and feature dimensions, see point 2 of global rebuttal above. Note that the image size used in DIFT and its competitor self-supervised feature is the same for each task, i.e., DIFT$\_{sd}$ vs. OpenCLIP, DIFT$\_{adm}$ vs. DINO, so the comparison should be fair. The per-pixel feature is extracted through bilinear interpolation on the feature map. We also include the code of DIFT (see `sd_featurizer.py` and `adm_featurizer.py`) in the submitted supplementary material.
>
> **Misclassification and missing comparisons in semantic matching**.
>
> Thanks for the correction and we will revise the tables accordingly and also include PWarpC as baseline.
>
> **Aggregation of multi-layer diffusion features**.
>
> For all the results in the paper, only single layer diffusion features are used. In our preliminary experiments, we also found that aggregating features from multiple layers could slightly improve the performance. However, this also inevitably introduces many design choices (e.g., how to do the aggregation) and hyper-parameters (e.g., the layer and time step t to aggregate and the weight on each of them), which could vary across different downstream tasks. Optimizing them could further improve the performance, but also entangles the quality of the features with the tuning of these design choices and hyper-parameters. Since the main focus of our paper is to demonstrate correspondence emerges from image diffusion without explicit supervision, we focus on the most simple technique and the raw off-the-shelf single timestep/layer diffusion features. We’ll leave how to improve the performance by constructing a multi-layer/timestep diffusion feature pyramid to future work.
>
> **Number of network inference passes used for each image**.
>
> It only takes one single network pass when extracting DIFT for each image. As mentioned in Line 128-130, to enhance the stability of the representation in the presence of random noise added to the input image, we extract features from multiple noisy versions with different samples of noise, and average them to form the final representation. As mentioned in the first paragraph (Line 470-472) of Appendix C, this is usually being done in a batch manner so only one network inference is needed per image (see the `forward()` function in `sd_featurizer.py` and `adm_featurizer.py` of the code in our submitted supplementary material). We usually sample 8 noise per image but sometimes shrink it to 4 when having memory issues. Changing from 8 to 1 noise per image only lead to very small performance drop, e.g., on SPair-71k, DIFT$\_{sd}$: PCK 59.5&rarr;57.9; DIFT$\_{adm}$: PCK 52.0&rarr;51.1.

---

> ### Comment · Area_Chair_Un5A · 2023-08-13
>
> Hi! It is now the discussion period. Please have a look at the rebuttal and let us know if it affected your thoughts on the paper in any way.

---

> > ### Comment · Reviewer_rXbs · 2023-08-16
> >
> > The authors addressed my concerns. I will upgrade to accept.

---

> > > ### Author Response · Authors · 2023-08-21
> > >
> > > We're thankful that you've taken the time to read our response and raise the score! Your feedback is highly treasured, and we'll make the necessary revisions to our paper in its future version.

---

### Official Review · Reviewer_zLC3 · 2023-06-30

**Soundness:** 2 fair
**Presentation:** 3 good
**Contribution:** 3 good
**Rating:** 6
**Confidence:** 5

**Summary:**

The paper proposes to use off-the-shelf generative networks based on denoising diffusion models to find local correspondences. The paper is extremely simple: instead of generating samples purely from random noise or doing some kind of image-based conditioning, the method just adds random noise to the input image and takes some intermediate level in the denoising U-Net as a dense feature map, from which it can extract sparse features. Feature matching is done via cosine distances. Keypoints may be provided (e.g. for semantic matching) or taken with an off-the-shelf keypoint detector, such as SuperPoint. The authors show that their approach works for high-level semantics (e.g. an eye across different animals or even species) and geometric correspondences, and can also track points across time in video sequences. Parameters such as the stage in the denoising process or which intermediate layers to use are chosen per dataset. There are some tricks (e.g. featuremaps are averaged over batches with different noise inputs), but that's about it.

**Strengths:**

1. Cool idea.
2. Very simple. The actual method section is a single paragraph and doesn't require any math. The paper is well written and easy to understand.
3. Very good results on multiple tasks, including semantic matching and tracking in video sequences, without training or fine-tuning.

**Weaknesses:**

1. My main complaint is that I find it hard to believe this approach will work well across arbitrary geometric changes. I have experience in this field and conclusions drawn from HPatches rarely translate to real scenarios. The paper gives very few results, which makes these experiments unreliable and difficult to trust (my guess is that this gap will disappear when the method is properly evaluated). For instance:
- What is the image size?
- How many points do you use?
- Do you use both the illumination and viewpoint splits for HPatches (the supplementary material suggests so)? If so, why not split the results by sequence type?
- Why not use the more standard MMA metric from D2-Net (which isn't perfect but is easier to understand), instead of estimating homographies? Why not both?
- Why not use modern baselines? The most recent method is from 2019 (R2D2): see e.g. DISK, PoSFeat, ALIKE, or SILK (citations below). (Note: CAPS is weakly supervised and works worse than the DISK variant supervised only with epipolar geometry, as far as I know.)
- Could you provide a simple precision/recall curve and compare it against traditional local feature methods? My intuition is that it would be worse.
- What happens if you add RANSAC?

I generally discount claims on this dataset and steer people towards evaluation benchmarks focusing on downstream tasks, such as visualocalization.net or the image matching challenge (https://image-matching-workshop.github.io/). I understand that this is likely beyond the scope of this paper, and I think it would be acceptable that it just shows these features work reasonably well for rigid matching, but if you want to claim that "though not trained using any explicit geometry supervision, DIFT outperforms prior state-of-the-art methods that utilize explicit geometric supervision signals designed specifically for this task, such as correspondences obtained from Structure from Motion pipelines" (L210-213), then you're going to have to substantiate that much better.

2. Given that the paper is so simple and the concept itself is very easy to explain, it would be nice to see more introspection. For instance, showing qualitative results using PCA or t-SNE to cluster the features (see for instance the videos from DINO v2) would help understand what's going on.

3. Important details are relegated to the supplementary material:
- The amount of noise used per dataset. It's surprising that, for instance, for geometric correspondence with Stable Diffusion the best results are with t=0, which means basically no noise (if I understand the formulation correctly; I looked at the code).
- The fact that for semantic correspondence per-class prompts are used. I think this is a reasonable assumption, but it should be mentioned. It also suggests the method may not work so well in non-object-centric images.

**Questions:**

The paper is cool and novel, but given the issues above I'll mark it as borderline (BA, since I only have BA and BR) and wait on the rebuttal and reviewer discussion.

- How do you deal with the loss in resolution, especially for geometric matching? The supplementary material says the method uses the 11th upsampling block out of 18 for ADM, and the 2nd block out of 4 for SD. Do you simply upscale the feature maps? And if so how do you achieve accurately localized correspondences?
- Do you choose per-dataset hyperparameters on a validation set or the test set?
- What is the computational cost and how does it compare with other methods?
- What does "after removing outliers" mean in fig 10 in the supplementary material? Why not show the raw matches and color-code them according to correct/incorrect?
- What happens without per-class prompts?

Notes:

- It's nice to see that the method is sensitive to the choice of t but varies smoothly (Fig. 5). It would be nice to see this in other datasets, though. And what about n?
- The images in Fig. 10 in the supplementary material have the wrong colors (I guess BGR, from loading/saving with OpenCV).
- Please add the dataset and metric to the headers in Tables 1 and 2 (one's missing the metric, the other two the dataset names).

Citations:

[DISK] https://arxiv.org/abs/2006.13566 (https://github.com/cvlab-epfl/disk)
[PoSFeat] https://openaccess.thecvf.com/content/CVPR2022/papers/Li_Decoupling_Makes_Weakly_Supervised_Local_Feature_Better_CVPR_2022_paper.pdf (https://github.com/The-Learning-And-Vision-Atelier-LAVA/PoSFeat)
[ALIKE] https://arxiv.org/pdf/2112.02906.pdf (https://github.com/Shiaoming/ALIKE)
[SILK] https://arxiv.org/abs/2304.06194 (https://github.com/facebookresearch/silk)

---

> ### Author Rebuttal · Authors · 2023-08-06
>
> Thank you for the insightful feedback! Please see our response below:
>
> **Implementation details and evaluations on HPatches**.
> - Image size: all images are resized to 768x768 then fed into the network to extract feature maps.
> - Number of points: following CAPS, we use SuperPoint to extract keypoints and it has 1.7k points per image on average.
> - MMA metric: we plot MMA curves in Fig. 4 of the attached pdf.
> - Modern baselines: We'll include the suggested methods in the revised comparison table. We include the DISK variant trained with epipolar supervision only (DISK$\_{epi}$) in the MMA figure above and report their homography numbers below.
> - Dataset splits: Tab. 3 shows the overall accuracy. Here’re the per-split numbers (also for DISK$\_{epi}$):
> | Method | $\epsilon=1$ | $\epsilon=3$ | $\epsilon=5$ |
> | ----- | ------ | ----- | ----- |
> | Illumination |  | | |
> | DIFT$\_{sd}$ | 64.2 |  93.5 |  97.3 |
> | DISK$\_{epi}$ | 66.9 |  93.1 | 96.9 |
> | Viewpoint | | | |
> | DIFT$\_{sd}$ | 28.9 |  63.2 | 72.9 |
> | DISK$\_{epi}$ | 28.6 | 62.1|  75 |
> - PR curve comparison with traditional local feature methods: using the same SIFT keypoints, we compare the mean Average Precision (mAP) of all image pairs using different feature descriptors with a threshold of 5 pixels: SIFT=74.9, DIFT$\_{sd}$=69.7, DINO=45.6. Although both DIFT and SIFT have high precision at high recall, the mAP for DIFT is lower because DIFT's scoring function is not optimized. DIFT achieves better MMA and homography accuracy than SIFT.
> - Add RANSAC: given a pair of images, we first use cosine distance to find mutual nearest neighbor matches, then use `cv2.findHomography()` with `method=cv2.LMEDS` to remove outliers and calculate the homography transformation. Empirically, we find LMEDS works better than RANSAC.
> - Loss of resolution: after getting the dense (but also lower-resolution) feature map, pixel-level features are extracted via bilinear interpolation. We use SuperPoint to localize the keypoints then extract their DIFT features to do matching. More details about image/feature map resolution are in point 2 of the global rebuttal.
>
> To address the reviewer's concern, we will modify our claim: "though not trained with any explicit geometry supervision, DIFT still achieves competitive performance on HPatches compared to the methods explicitly trained for geometric correspondence with weak/epipolar supervision".
>
> **More introspection of DIFT**.
>
> Please refer to point 1 of the global rebuttal above, and Fig. 1 and 2 of the attached pdf.
> - PCA visualization: we visualize the first 3 PCA components of DIFT$\_{sd}$ on segmented object pairs. As shown in Fig. 1, the object parts that share the same semantic meaning tend to have similar DIFT embeddings.
> - Different t for different correspondence tasks: we visualize the predicted clean images with different time step t in Fig. 2 as an intuitive way to demonstrate that features contains more semantic-level information and less low-level details with the increase of t, which partially validate the usage of very small t for geometric correspondence.
>
> **The usage of per-class prompt in semantic correspondence**.
>
> Changing per-class prompt to a null prompt (empty string) will only lead a very small performance drop for DIFT$\_{sd}$ on SPair-71k: 59.5&rarr;57.6 as in Tab. 1 of the main paper.
>
> We'll mention the usage of per-class prompt in the revised main paper. Note that only DIFT$\_{sd}$ needs a prompt as part of the input while DIFT$\_{adm}$ doesn’t. This per-class prompt is only used in semantic correspondence tasks, while null prompt is used for the others. The prompt design is simple and not image specific, only using class name, i.e., “a photo of a {class}”. For CUB, we actually use the same prompt for all images: “a photo of a bird”.
>
> **Wrong color format in Fig. 10; Missing headers in Tab. 1 and 2; Important details in the supp**.
>
> We apologize for the error. We'll fix this and table headers in the revision. We'll also add these implementation details in the main text plus a pointer to the appendix.
>
> **Per-dataset hyperparameters**.
>
> Most of the datasets used in the paper don't have a validation set, so we chose the two hyperparameters of DIFT (i.e., time step t and block index n) based on the test performance. But since our hyperparameter search on DIFT is pretty coarse, we do not believe our results are due to overfitting. Fig. 5 in the main paper and Fig. 5, 6 and 7 in the attached pdf show the performance for all possible $t$ and $n$ values; we will show similar plots for all tasks in the camera ready supplementary.  Note that, for fair comparison, we also densely grid search which layer to extract feature for DINO and OpenCLIP for each dataset. Also as mentioned in Line 154-156, for semantic correspondence, the hyperparameter tuning is only done on SPair and fixed afterwards on other datasets.
>
> **Computation cost**.
>
> Please refer to point 2 of global rebuttal above.
>
> Briefly speaking, DIFT takes similar running time as its competitor self-supervised features, e.g., on one single A6000, for each image, DIFT$\_{sd}$ takes 203 ms vs. OpenCLIP’s 231 ms. We'll also include these details in the revision.
>
> **"after removing outliers" in Fig. 10**.
>
> We only visualize the matches that are marked as "inliner" after applying `cv2.findHomography()`. We'll add figures with colored raw matches in the revision.
>
> **Ablation on t and n**.
>
> We ablate DIFT$\_{sd}$ with different t on HPatches and DAVIS, as in Fig. 5 and 6 of the attached pdf, where similarly to Fig. 5 of the paper, performance varies across t smoothly.
>
> We also ablate which layer to extract DIFT$\_{sd}$ for SPair in Fig. 7 of the attached pdf, where layer index i is more fine-grained than block index n, i.e., SD U-Net has 4 upsampling blocks with 15 layers in total. The transition from n to i is 0/1/2/3 to 3/7/11/14 (both start from 0). We can see the performance varies but there's a wide range choice of i having good performance.

---

> > ### Comment · Reviewer_zLC3 · 2023-08-16
> >
> > Woops! I just noticed that I pasted things in the wrong fields when copying the text from my editor to openreview, and the summary twice (so things were shifted by one field). Sorry, I try to be careful with these things but I had many papers to review. I guess it was easy to understand and all the content was there, so no harm done. I have fixed this now.
> >
> > **[Experimental details on HPatches]**
> >
> > Thanks for clarifying this. I ask for these details because I've seen papers do very questionable things (e.g. resizing images to 320x240 and benchmarking on that, or running RANSAC with horrible defaults). The comparison with DISK is convincing.
> >
> > At first sight it's a bit surprising that the performance for DIFT_{sd} is quite good at low pixel thresholds when the feature maps at are 1/8 the input resolution, but SuperPoint does the same for the feature maps, and the keypoints are extracted at the input resolution, so it makes sense. (You might have more issues discriminating the features with tighter non-maxima-suppression.)
> >
> > I find it surprising than LMEDS works better than RANSAC (my experience is 100% the opposite), and this very likely won't hold on more challenging datasets. It is also possible that the authors are not tuning RANSAC well enough (it's difficult to do this well -- plus OpenCV 4 has newer RANSAC variants built-in that perform quite a bit better than vanilla RANSAC), but maybe HPatches is just easy enough and it doesn't make much of a difference.
> >
> > **To address the reviewer's concern, we will modify our claim: "though not trained with any explicit geometry supervision, DIFT still achieves competitive performance on HPatches compared to the methods explicitly trained for geometric correspondence with weak/epipolar supervision".**
> >
> > To clarify, my concern was not that you claim that the method works well on HPatches as much as on "geometric correspondence". HPatches is in my opinion not a good dataset to evaluate geometric correspondence, because it only has homographies (and half of it illumination changes, which is an even more specific problem). And I say this knowing that that's what many papers on geometric matching do (which I often reject, because again, it's not a representative problem).
> >
> > The fact that the method works this well, off-the-shelf, on HPatches is really cool, and my reception of the paper might have been warmer if you had only included the other experiments. But I'm not going to penalize the paper for that. While I encourage you to acknowledge the limitations of the dataset, I'll raise my score to "weak accept" now that I can trust the evaluation, given the details the authors provided in the rebuttal.
> >
> > **[Ablations on t on other datasets, and on n]**
> >
> > Thanks, I think this material strengthens the paper, particularly since there's the focus of the paper is demonstrating how to use these off-the-shelf models for a different purpose (same reason why I think it's important to note the prompt, and what happens when you leave it blank). Consider my questions answered.

---

> > > ### Author Response · Authors · 2023-08-21
> > >
> > > Thanks a lot for reading our rebuttal and raising the score! We really appreciate your valuable feedback and suggestions especially on the geometric correspondence section. We'll revise our paper accordingly in the future version.

---

> ### Comment · Area_Chair_Un5A · 2023-08-13
>
> Hi! It is now the discussion period. Please have a look at the rebuttal and let us know if it affected your thoughts on the paper in any way.

---

### Official Review · Reviewer_4nEG · 2023-07-06

**Soundness:** 3 good
**Presentation:** 4 excellent
**Contribution:** 3 good
**Rating:** 7
**Confidence:** 4

**Summary:**

The paper addresses a classical computer vision problem, ie, points correspondence. The authors show that the feature maps of the decoder of a diffusion model U-Net enable robust feature matching with a simple nearest neighbor search. Semantic or geometric correspondence can be achieved, by selecting the appropriate denoiser time step. The authors report results, quantitative analysis and comparison on several benchmarks, chosen for each specific task.

**Strengths:**

The authors observe empirically that the features contained in the U-Net decoder are powerful features  for image correspondence. They also show that those features embed different levels of semantic information, depending on the time step of the denoiser.

It is particularly relevant given the fact that diffusion models used by the authors are pretrained (the authors only add noise to the input image), and that the drawn conclusions are consistent over different DMs.

The set of experiments reported in the paper are thorough and results are convincing. The variety of the results, quantitative and qualitative, demonstrate the reliability and flexibility of such features.

I particularly enjoyed reading a paper presented with such simplicity and clarity.

**Weaknesses:**

The drawback of the approach is probably the computational complexity that is intrinsic to any point matching approach relying on nearest neighbor search based on high dimensional features.

It is not clear from the paper (nor the supplementary material), if choosing the optimal time denoising time step is critical or not. A corollary of this question would be: do we expect the semantic level contained in the features to degrade smoothly from large time step to small ones?

**Questions:**

1) In figure 3,  it seems that the semantic information is such that the right ear feature is distinct from the left ear one. Is it a general case or does this happen in this specific example only due to a very similar pose in the source and the target image?

2) Did the authors try to apply clustering based on U-Net features?

3) Could the authors elaborate on the  computational time. What is the dimension of a DIFT feature?

**Limitations:**

the authors rightly mentioned the ethical issues related to generative models .

---

> ### Author Rebuttal · Authors · 2023-08-08
>
> Thank you for the insightful feedback! Please find our response as below:
>
> **Clustering on U-Net features**.
>
> Please refer to point 1 of the global rebuttal above and Fig.1 of the attached pdf, where we we visualize the first three PCA components of DIFT$\_{sd}$ on the segmented instance pairs and we can see that the same kind of object parts share similar feature embeddings. We'll include more such visualization in the future version.
>
> **Computation time and feature dimensions**.
>
> Please refer to point 2 and 3 of the global rebuttal above.
>
> Breifly speaking, DIFT takes similar running time as its competitor self-supervised features, e.g., on one single A6000, for each image, DIFT$\_{sd}$ takes 203 ms vs. OpenCLIP’s 231 ms, DIFT$\_{adm}$ takes 110 ms vs. DINO’s 154 ms. The feature dimension depends on which U-Net layer is extracted from, and we list the details for every correspondence tasks above.
>
> We'll also include these details in the future version of our paper.
>
> **Is the choice of time step t critial?**.
>
> In the main paper, we briefly discussed the influence of time step t in Sec. 5.2 Line 176-178 and Fig. 5 shows how the semantic correspondence performance varies across different t. It can be seen that curve is pretty smooth and there’s a large range of t where DIFT gives competitive performance. But the accuracy also decreases significantly when t becomes too large or too small.
>
> As suggested by reviewer zLC3, we have added new experiments that ablate the choice of $t$ on the geometric correspondence (HPatches) and temporal correspondence (DAVIS) tasks in Fig. 5 and 6 of the rebuttal document respectively, where they share the similar observation as above.
>
> Given these experiments,  we would say the choice of t is definitely an important hyperparameter that user need to tune for specific downstream correspondence tasks, but the trend is quite smooth and the change is also not sharp, so it’s not hard to find a good enough t within a reasonable number of trials. Based on our experiments, intuitively we find that within a reasonable range (e.g., t cannot be too large, otherwise $x_t$ is too noisy), larger t usually leads to more semantic-level features and smaller t leads to features that contain more low-level image details. Users can also use this heuristics to help narrow down the search.
>
> In addition to these new ablations and visualizations, Figure 2 in the rebuttal document attempts to give an intuition for the effect of $t$. Please see point 1 of the global rebuttal for further explanations of this figure.
>
> **Do we expect the semantic level contained in the features to degrade smoothly from large time step to small ones?**
>
> As mentioned in the above comments, based on the ablations and experiments we have, we think this is a good heuristics to have when tuning the time step t for downstream tasks. The smoothness could also be verified in the low-sensitivity of semantic correspondence performance to the choice of t, as in Fig. 5 of the main paper. In Fig. 2 of the attached pdf and point 1 of the global rebuttal above, we also give an intuitive explanation on why this would happen by visualizing the predicted clean image at different time step t.
>
> **Distinction between symmetric object parts using DIFT**.
>
> There’re a lot of such symmetric keypoints In SPair-71k/CUB/WILLOW (e.g., left vs. right ear/handler/wing, fore-wheel vs. back-wheel) and the prediction has to distinguish them from each other in order to be counted as correct. So the high accuracy numbers on these benchmarks actually indicate that DIFT is able to handle these cases pretty well in general. In addition to Fig. 3, the last two rows of Fig. 4 also demonstrate a few cross-category cases on distinguishing left/right paws/eyes. Especially in the last row, we can see even the pose between source and target are quite different, DIFT with simple cosine similarities is still able to give correct predictions.
>
> **Large computational complexity that is intrinsic to any point matching approach relying on nearest neighbor search on high dimensional features**.
>
> The main focus of our paper is to demonstrate that correspondence emerges from image diffusion models without explicit supervision. In order to show that, we use simple technique to extract the feature representation inside U-Net, paired with the most straightforward feature matching strategy (i.e., nearest neighbor using cosine distance) and it achieves competitive performance on several correspondence tasks. The simplicity of the matching mechanism actually further proves the good quality of diffusion features themselves.
>
> But we agree that this nearest neighbor matching strategy is definitely not ideal, and more sophisticated designs from the feature matching literature could further boost DIFT’s performance and efficiency, e.g., a coarse-to-fine matching strategy. Meanwhile, in terms of efficiency in practice, note that DIFT is basically as fast as other off-the-shelf self-supervised features (e.g., DIFT$\_{sd}$ takes 203 ms running on a single input image, see more details in point 2 of the global rebuttal above). And when calculating cosine distance, we could L2-normalize each feature independently and then dot product each other, which is also very efficient in terms of time and memory, compared to other correspondence methods [27, 31] that use the attention mechanism to calculate the correlation. In addition, since DIFT feature is image-independent so we can pre-calculate it for each image and then save them in advance before doing the matching, whereas some methods [27, 31] that require a pair of images as network input cannot take advantage of this pre-processing, especially when the same image need to be queried multiple times.
>
> We'll add this clarification into the future version of our paper.

---

> > ### Comment · Reviewer_4nEG · 2023-08-21
> > **Official comment by reviewer 4nEG**
> >
> > Thanks to the authors for the detailed rebuttal and clarifications regarding experimental setup. I believe that these clarifications, together with the additional experiments, will consolidate the paper. My concerns have been addressed and I maintain a recommendation for accept.

---

> > > ### Author Response · Authors · 2023-08-21
> > >
> > > Thank you for investing time in reading our rebuttal and for responding with a higher score! Your feedback is truly cherished, and we'll revise our paper accordingly in the future version.

---

> ### Comment · Area_Chair_Un5A · 2023-08-13
>
> Hi! It is now the discussion period. Please have a look at the rebuttal and let us know if it affected your thoughts on the paper in any way.

---

### Official Review · Reviewer_u3f5 · 2023-07-07

**Soundness:** 2 fair
**Presentation:** 3 good
**Contribution:** 4 excellent
**Rating:** 7
**Confidence:** 5

**Summary:**

This paper introduces DIFT, a method to yield emergent correspondence from image diffusion models without training or additional fine-tuning.
The method is simple - given an image (or an image pair), DIFT adds noise to the image to simulate the forward diffusion process, and pass it to the U-Net of a pretrained diffusion model to extract feature maps.
The authors discover that by simply computing the cosine similarity between the emergent feature maps, one can establish strong semantic, geometric and temporal correspondences without training.

**Strengths:**

* The discovery of emergent correspondences from image diffusion is novel. The proposed method is surprisingly simple as well, and this opens new possibilities and research directions for future work.

* Strong performances on standard benchmarks of semantic correspondence, geometric correspondence and temporal correspondence.

* The writing is clear and easy to follow.

**Weaknesses:**

* Missing evaluation of PF-PASCAL for semantic correspondence. This is not a critical drawback, as results on PF-PASCAL tend to be saturated.

* Incomplete implementation details. What was the image size / feature map size used to establish the correspondences? Image size is a critical factor in many image correspondence methods.

* Lack of latency and computation analysis. This is crucial to identify the applicability of DIFT to real world scenarios.

* Lack of rationale or analysis on how exactly the image diffusion models can yield such feature maps easily. While some motivation is provided in the introduction, it seems insufficient for the readers to fully understand how the correspondences are emerging from image diffusion. Analyzing some failure cases could be helpful.

**Questions:**

Please refer to the weaknesses section. The idea, simplicity, and novelty of the paper is strong, but I believe the paper lacks the experimental details and analyses to be accepted as-is.

**Limitations:**

The authors have included the ethical considerations in the paper, but not the limitations.

---

> ### Author Rebuttal · Authors · 2023-08-05
>
> Thank you for the insightful feedback! Please find our response as below:
>
> **Missing evaluation on PF-PASCAL**:
>
> Here’re the comparison on PF-PASCAL:
> | Method | PCK@$\alpha_{img}$=0.1 |
> | ----- | ----- |
> | PWarpC [Truong et al. CVPR 2022] (see review by rXbs) |  87.6 |
> | DIFT$\_{sd}$ | 84.6|
> | OpenCLIP | 61.1 |
> | DIFT$\_{adm}$ | 72.5 |
> | DINO | 53.6 |
>
>
>  PWarpC is a state-of-the-art weakly-supervised method mentioned by Reviewer rXbs, trained on training images with only image-level class labels. We didn’t do evaluation on PF-PASCAL at the beginning because we found over half of (i.e., 302 out of 506) the test images actually also appear in the training set, which makes the benchmark numbers much less convincing, and also partially explains why the previous supervised methods tend to have much higher test accuracy on PF-PASCAL vs. PF-WILLOW (e.g., over 90 vs. around 70) even using exactly the same trained model. And this duplication issue of train/test images also gives huge disadvantage to the methods that are never adapted (either supervised or unsupervised) on the training set before evaluation. However, even in this case, DIFT still demonstrates competitive performance as well as huge gains vs. other off-the-shelf self-supervised features. We’ll add this clarification in the future version of the paper.
>
> **Latency of DIFT**:
>
> Please refer to point 3 of the global rebuttal above.
>
> Breifly speaking, DIFT takes similar running time as its competitor self-supervised features, e.g., on one single A6000, for each image, DIFT$\_{sd}$ takes 203 ms vs. OpenCLIP’s 231 ms, DIFT$\_{adm}$ takes 110 ms vs. DINO’s 154 ms. We'll include these details in the future version of the paper.
>
> **Lack of experiment details, e.g., image size, feature map size**.
>
> The input image size varies across different tasks but we always keep it the same within the comparison vs. other off-the-shelf self-supervised features (i.e., DIFT$\_{adm}$ vs. DINO, DIFT$\_{sd}$ vs. OpenCLIP) thus the comparison should be fair. For every correspondence tasks, we include the details of image size and feature map size/dimension in the point 2 of global rebuttal above. We also included implementation details in Appendix C and the code of DIFT (see `sd_featurizer.py` and `adm_featurizer.py`) in the submitted supplementary material. We'll include above details either in the main paper or add a pointer to the supplementary.
>
> **Lack of analysis on why correspondence emerges from image diffusion**.
>
> Please refer to point 1 of the global rebuttal above.
>
> We provide two more visualizations in the attached pdf to help audience better understand how DIFT works. In Fig. 1, we visualize the first three PCA components of DIFT$\_{sd}$ on the segmented instance pairs and we can see that the same kind of object parts share similar features. In Fig. 2, we visualize the predicted clean images at different time step $t$ and we can see that, within a reasonable range, when $t$ gets larger, the predicted clean images remain the overall structure but have less details, suggesting DIFT contains more semantic-level information and less low-level features with the increase of $t$. It also explains that we need a small $t$ for correspondence tasks that require details and relatively large $t$ for semantic correspondence.
>
> We believe that the diffusion training objective (i.e., coarse-to-fine reconstruction loss) requires the model to produce good, informative features for every pixel. This is in contrast to DIFT outperforms DINO and OpenCLIP that use image-level contrastive learning objectives. In our experiments, we have attempted to evaluate the importance of the training objective by specifically comparing DIFT$\_{adm}$ and DINO in all our evaluations: two models that share exactly the same training data, i.e., ImageNet-1k without labels.
>
> We'll include these figures and analysis in the future version of the paper.
>
> **Analyzing failure cases**.
>
> Please refer to Fig. 3 in the attached rebuttal pdf file for the failure case visualization, where we randomly select two examples from the SPair-71k category with the lowest per-class accuracy (i.e., boat and bottle). We can see that, when the semantic definition of key points are ambiguous or the appearance change between source and target images are too dramatic, DIFT$_{sd}$ fails to predict the ground-truth labels. We'll include more visualizations and analysis on different failure cases in the future version of the paper.

---

> > ### Comment · Reviewer_u3f5 · 2023-08-19
> >
> > I appreciate the authors for their detailed responses to my concerns. I believe that by reflecting these responses (and the responses to the other reviewers' concerns), the manuscript will be substantially improved. I would like to improve my rating to accept.

---

> > > ### Author Response · Authors · 2023-08-21
> > >
> > > Thanks for reading our rebuttal and getting back to us with a higher score! We appreciate the valuable feedbacks you have given and we will revise our paper accordingly in the future version.

---

> ### Comment · Area_Chair_Un5A · 2023-08-13
>
> Hi! It is now the discussion period. Please have a look at the rebuttal and let us know if it affected your thoughts on the paper in any way.

---

### Official Review · Reviewer_4Vxk · 2023-07-07

**Soundness:** 3 good
**Presentation:** 3 good
**Contribution:** 3 good
**Rating:** 4
**Confidence:** 5

**Summary:**

The authors proposed a method for semantic correspondence using pretrained diffusion model as a feature extractor of the images. Without explicitly training on the additional data/annotations, a simple feature matching based on winner-take-all strategy with cosine distance metric surpasses the previous works on three different tasks; semantic/geometric/temporal correspondence.

**Strengths:**

- Exploration on the usage of diffusion model for correspondence tasks
- Good performances just with frozen pretrained model & simple matching pipeline

**Weaknesses:**

1. Limited novelty
- Merely replacing the backbone network for feature extraction with the latest model in a straightforward manner does not captivate my interest. It also fails to provide new insights to the readers. It would be beneficial to delve deeper into why these pretrained diffusion models outperform previous backbones like DINO, CLIP, or ResNet.

2. Limited demonstrations
- The paper lacks several essential ablation studies that explore architectural design choices. These studies could include comparing models with and without a decoder, evaluating different options for Q/K/V selection, and assessing the impact of finetuning on correspondence datasets.
- Given the relatively low originality, the paper would benefit from showcasing more diverse pixel-level prediction tasks. This could involve demonstrating the model's performance in object detection, image/ video segmentation, as well as other correspondence tasks such as depth estimation and optical flow. These additional demonstrations would significantly strengthen the paper.

**Questions:**

It would be interesting to see how the performances can be pushed further when sophisticated optimization techniques are applied to the raw matching similarities d(f^1,f^2).

**Limitations:**

The limitations and potential societal impact of this work are appropriately discussed

---

> ### Author Rebuttal · Authors · 2023-08-05
>
> Thank you for the insightful feedback! Please find our response as below:
>
> **Limited novelty**.
>
> We would like to point out that the "backbone" we are proposing to use comes from a diffusion model which is trained with a generative modeling objective that has prima facie little to do with learning good features. Thus the fact that correspondence emerges from this generative training and can even outperform techniques specifically designed for representation learning (DINO, OpenCLIP) is a surprising finding. Our simple approach for extracting this correspondence suggests that this is something intrinsic to the diffusion training objective. We think it's valuable to share this evidence with the community so that more interesting research directions could be explored, such as understanding why diffusion objectives are more suitable for capturing correspondence, and perhaps rethinking the diffusion model as a self-supervised learner.
>
> **Explain why pre-trained diffusion models outperforms previous backbones like DINO, OpenCLIP**.
>
> We agree that more insights would be useful. As mentioned in the global rebuttal (point 1), we have added two new visualizations that we hope can shed some light on the features. We also have a conjecture that the diffusion model training objective (i.e., coarse-to-fine reconstruction loss) requires the model to produce good, informative features for every pixel compared to contrastive learning objectives (i.e., loss on image-level global feature).  In our experiments, we have attempted to evaluate the importance of the training objective by specifically comparing DIFT$\_{adm}$ and DINO in all our evaluations, where they share exactly the same training data, i.e., ImageNet-1k without labels. We will add this discussion to the camera ready.
>
> **Evaluating more pixel-level prediction tasks**.
>
> We indeed evaluate on video segmentation (DAVIS) and human keypoint tracking (JHMDB) in Sec. 6.2 and DIFT shows competitive performance although never being trained on such supervision or even video data. We have now also evaluated on image segmentation, where we freeze the backbone and only train a linear head on top of different feature extractors.
> The mIOU results on ADE20k val set (single-scale evaluation)  are as follows:
>
> | Method | mIOU |
> | --------- | -------- |
> | DIFT$\_{adm}$ | 29.2 |
> | DIFT$\_{sd}$ | 34.5 |
> | DINO | 31.1 |
> | OpenCLIP | 40.9 |
>
> Note that this task is different from correspondence tasks with feature matching because it requires training a head with downstream supervision thus introduces additional parameters and hyperparameters. We performed an additional experiment where we evaluated the off-the-shelf feature correspondence using COCO image segmentation labels (as a precursor to object discovery; see Fig. 3 in STEGO [20]):
>
> | Method | AP |
> | --------- | -------- |
> | DIFT$\_{adm}$ | 80 |
> | DIFT$\_{sd}$ | 78 |
> | DINO | 79 |
> | OpenCLIP | 65 |
>
> DIFT achieves competitive performance on both these tasks.
>
> Unfortunately for depth estimation and optical flow, current pipelines (e.g., RAFT) usually involve a complex global optimization that is beyond simple local feature matching.
>
> **Exploring more sophisticated design choices or optimization techniques to further improve the correspondence performance**.
>
> In our preliminary experiments, we do find that more sophisticated task-specific adaptation mechanism could further enhance the diffusion features, e.g., finetuning the network with specific downstream correspondence supervision. However, this would inevitably entangle the quality of the features themselves with the efficacy/strategy/hyperparameters of the adaptation procedure. To keep the focus on the evaluation of the representation itself, we chose to avoid any additional adaptation to demonstrate the raw quality of the off-the-shelf DIFT. Also see the discussion (Line 458-464) in Appendix B.
>
> “Comparing models with and without a decoder”: if "decoder" means the VAE decoder of the diffusion model, we indeed compared two types of diffusion models in all the evaluations: SD is a latent diffusion model with VAE decoder while ADM is directly on pixel space without en/de-coders. But note that for DIFT$\_{sd}$, the decoder is actually not used during inference; if “decoder” means whether to attach a task-specific decoder for downstream correspondence tasks, we chose not to do that because it will require finetuning with supervision, see the discussion above.
>
> “Evaluating different options for Q/K/V selection”: our current implementation follows the most straight-forward way to get the intermediate feature map after each resnet-attention upsampling blocks inside U-Net (see the class `MyUNet2DConditionModel` of `sd_featurizer.py` in our submitted code). There contains multiple self-attention and cross-attention layers within each SD U-Net attention block. Delving into it and exploring key/value/query tokens as representation could potentially give performance boost, but also complicates the design choice and usage (different tasks require different set of hyperparameters).

---

> ### Comment · Area_Chair_Un5A · 2023-08-13
>
> It is now the discussion period! Please have a look at the rebuttal and let us know if it affected your thoughts on the paper in any way!

---

### Author Rebuttal · Authors · 2023-08-05

We thank all the reviewers for their time and valuable feedbacks! We are encouraged that reviewers find that our paper is well-written and easy to follow (u3f5, 4nEG, zLC3, rXbs), that our approach achieves good performance with a simple technique (all 5 reviewers!) along with cool and novel ideas (u3f5, zLC3). Here, we answer a few global questions below:

**1. Understand why correspondence emerges from image diffusion, and why it outperforms previous backbones like DINO, OpenCLIP**.

We agree that understanding why correspondence emerges is an intriguing question. Inspired by reviewers' suggestions, we add two more visualizations in the attached pdf. In Fig. 1, for each pair of images, we extract DIFT$\_{sd}$ from the segmented instances, then compute PCA and visualize the first 3 components, where each component serves as a color channel. We can see the same object parts share similar embeddings, which also demonstrates the emergent correspondence. To further explore how this correspondence depends on $t$, in Fig. 2, for the same clean image, we first add different amount of noise to get different $x_t$ following Eq. 1, then feed it into network $\epsilon_\theta$ together with time step $t$ to get the predicted clean image $\hat{x}_0^t=\frac{x_t - (\sqrt{1 - \alpha_t}) \epsilon
_\theta (x_t, t)}{\sqrt{\alpha_t}}$. We can see that, with the increase of $t$, more and more details are removed and only semantic-level features are preserved, and when $t$ becomes too large, even the object structure is distorted. Intuitively, this explains why we need a small $t$ for correspondences that requires details and a relatively large $t$ for semantic correspondence. We'll include more examples of both  visualizations in the future version.

As a more general explanation for the emergent correspondence, we have the following conjecture. We believe that the diffusion training objective (i.e., coarse-to-fine reconstruction loss) requires the model to produce good, informative features for every pixel. This is in contrast to DIFT outperforms DINO and OpenCLIP that use image-level contrastive learning objectives. In our experiments, we have attempted to evaluate the importance of the training objective by specifically comparing DIFT$\_{adm}$ and DINO in all our evaluations: two models that share exactly the same training data, i.e., ImageNet-1k without labels. A rigorous evaluation of our conjecture would be very interesting, but we lack the tools to perform such rigorous analysis: how to understand and explain why certain properties emerge from deep neural networks trained on large-scale data remains an unsolved research problem. We will add this discussion to the camera ready.

**2. Input image size, feature map size and feature dimension used in each correspondence task**.

The input image size varies across different tasks but we always keep it the same within the comparison vs. other off-the-shelf self-supervised features (i.e., DIFT$\_{adm}$ vs. DINO, DIFT$\_{sd}$ vs. OpenCLIP) thus it should be fair. For DIFT, feature map size and dimension also depend on which U-Net layer features are extracted from. ADM’s U-Net has 18 upsampling blocks and SD has 4 upsamping blocks (the definition of blocks varies). Below for each tasks, we list all the image sizes, block index $n$ ($n$ starts from 0), the size and dimension of feature maps. The per-pixel feature is extracted through bilinear interpolation on the feature map.  The code of DIFT (see `sd_featurizer.py` and `adm_featurizer.py`) is in the supplementary.

For semantic correspondence tasks, we use the input image size of 512x512 for DIFT$\_{adm}$, 768x768 for DIFT$\_{sd}$. n=4 for DIFT$
_{adm}$ so feature map size is 1/16 of input and dimension is 1024. n=1 for DIFT$\_{sd}$ so feature map size is 1/16 of input and dimension is 1280.

For experiments on HPatches, input image size is 768x768 for both DIFT$\_{adm}$ and DIFT$\_{sd}$. n=11 for DIFT$\_{adm}$ so feature map size is 1/2 of input and dimension is 512. n=2 for DIFT$\_{sd}$ so feature map size is 1/8 and dimension is 640.

For experiments on DAVIS, we use the same original video frame size (480p version of DAVIS, specific size varies across different videos) as in DINO [5]’s implementation, for both DIFT$\_{adm}$ and DIFT$\_{sd}$. n=7 for DIFT$\_{adm}$ so feature map size is 1⁄8 of input and dimension is 512. n=2 for DIFT$\_{sd}$ so feature map size is 1⁄8 of input and dimension is 640. For experiments on JHMDB, following CRW [29]’s implementation, we resize each video frame’s smaller size to 320 and keep the original aspect ratio. n=5 for DIFT$\_{adm}$ so feature map size is 1⁄8 of input and dimension is 1024. n=2 for DIFT$\_{sd}$ so feature map size is 1⁄8 of input and dimension is 640.

We'll add the above details to the future version of the paper.

**3. Latency of DIFT vs. its competitor self-supervised features**.

Since we only perform a single inference step when extracting DIFT features, it actually takes similar running time compared to its competitor self-supervised features with the same input image size. Taking semantic correspondence with the above configuration for example, on one single A6000, for each image, DIFT$\_{sd}$ takes 203 ms vs. OpenCLIP’s 231 ms, DIFT$\_{adm}$ takes 110 ms vs. DINO’s 154 ms.

In practice, as mentioned in the Line 128-130 of Sec. 4.2, since there is randomness when extracting DIFT, we actually use a batch of random noise to get an averaged feature map for each image to slightly boost stability and performance, which would increase the running time shown above. But if computation is a bottleneck, one can remove this optimization at the cost of a tiny loss in performance: e.g., on SPair-71k, DIFT$\_{sd}$: PCK 59.5&rarr;57.9; DIFT$\_{adm}$: PCK 52.0&rarr;51.1.

We'll add the above clarification to the future version of the paper.

---

### Decision · Program_Chairs · 2023-09-21

**Decision:**

Accept (poster)

**Comment:**

The majority of the reviews were positive and there was one review that was negative initially. After the rebuttal phase, most reviewers became more optimistic about the paper with the provided clarifications and the newly added results. It is thus strongly recommended that the rebuttal is integrated within the camera-ready. The negative reviewer never returned, but the comment on novelty is subjective and weak considering how other reviewers were convinced of the significance of the finding. The criticism of the experimental results is also weak as it is based on the weak-novelty argument, and asks for re-training of various architectures that is tangent to the main paper's argument. The AC hence is more in line with the other reviewers and recommends accepting the paper.